# Lmx1a is a master regulator of the cortical hem

**Igor Y Iskusnykh[1]\*[†], Nikolai Fattakhov[1][†], Yiran Li[2], Laure Bihannic[2], Matthew K Kirchner[1], Ekaterina Y Steshina[1], Paul A Northcott[2], Victor V Chizhikov[1]\***

[1]Department of Anatomy and Neurobiology, University of Tennessee Health Science Center, Memphis, United States; [2]Department of Developmental Neurobiology, St. Jude Children's Research Hospital, Memphis, United States

**Abstract** Development of the nervous system depends on signaling centers – specialized cellular populations that produce secreted molecules to regulate neurogenesis in the neighboring neuroepithelium. In some cases, signaling center cells also differentiate to produce key types of neurons. The formation of a signaling center involves its induction, the maintenance of expression of its secreted molecules, and cell differentiation and migration events. How these distinct processes are coordinated during signaling center development remains unknown. By performing studies in mice, we show that Lmx1a acts as a master regulator to orchestrate the formation and function of the cortical hem (CH), a critical signaling center that controls hippocampus development. Lmx1a co-regulates CH induction, its Wnt signaling, and the differentiation and migration of CH-derived Cajal–Retzius neurons. Combining RNAseq, genetic, and rescue experiments, we identified major downstream genes that mediate distinct Lmx1a-dependent processes. Our work revealed that signaling centers in the mammalian brain employ master regulatory genes and established a framework for analyzing signaling center development.

**\*For correspondence:**
iiskusny@uthsc.edu (IYI);
vchizhik@uthsc.edu (VVC)

[†]These authors contributed equally to this work

**Competing interest:** The authors declare that no competing interests exist.

## Editor's evaluation

This study clearly demonstrates the important role of Lmx1a in collaboration with Wnt signaling in the development of medial cortical structures, including the dentate gyrus. The studies elegantly show the role of Lmx1a in the development of the cortical hem and provide important new insights into the signals controlling formation of this structure. It also provides helpful context for previous studies in this area.

## Introduction

In the vertebrate central nervous system (CNS), tissue patterning and subsequent neurogenesis are regulated by signaling centers – localized groups of cells that produce secreted molecules that regulate the development of neighboring cells (*Bielen et al., 2017*; *Cavodeassi and Houart, 2012*; *Manfrin et al., 2019*). Formation of a signaling center is a complex process that involves its induction, segregation of its cells from the adjacent tissue, and the initiation and maintenance of expression of secreted molecules that mediate its function. In many cases, cells that comprise signaling centers also differentiate into neurons that migrate to specific brain regions and regulate their morphogenesis and/or form neural circuits (*Kiecker and Lumsden, 2012*; *Subramanian et al., 2009a*). Currently, little is known about how different aspects of signaling center formation are coordinated during development. In vertebrates, development of several cell types or even organs depends on master regulatory genes – intrinsic factors that orchestrate multiple developmental processes, such as transcription

factors *Atoh1* in cerebellar granule cells or *Pax6* in the eye (*Baker et al., 2018*; *Graw, 2010*; *Klisch et al., 2011*; *Srivastava et al., 2013*). It is unknown, however, whether the development of signaling centers in the CNS involves master regulatory genes.

One of the key signaling centers in the mammalian CNS is the cortical hem (CH), which develops at the telencephalic dorsal midline, between the choroid plexus and cortical neuroepithelium (*Grove et al., 1998*; *Jones et al., 2019*; *Moore and Iulianella, 2021*; *Sindhu et al., 2019*). The CH is necessary and sufficient for the development of the hippocampus in adjacent neuroepithelium (*Mangale et al., 2008*). Loss of CH or its signaling molecule Wnt3a leads to the near-complete loss of the hippocampus, associated with reduced proliferation in the hippocampal primordium (*Lee et al., 2000*). Canonical Wnt signaling from the CH (involving Wnt3a and likely other Wnts) also promotes the formation of the transhilar glial scaffold, which guides the migration of neural progenitors from the dentate neuroepithelium (DNe) in the ventricular zone to the hippocampal dentate gyrus (DG) (*Zhou et al., 2004*; *Parichha et al., 2022*). In addition to Wnt signaling, CH gives rise to Cajal–Retzius (CR) cells, short-lived neurons that migrate to the hippocampal field, where they form the hippocampal fissure (HF), the sulcus that separates the DG from the CA1 field. CR cells also promote the formation of the transhilar glial scaffold (*Bielle et al., 2005*; *Causeret et al., 2021*; *Hodge et al., 2013*; *Meyer et al., 2004*).

To date, intrinsic factors that confer CH fate remain elusive. For example, loss of transcription factors Dmrt3/4/5 or Gli3 compromises CH formation (*De Clercq et al., 2018*; *Fotaki et al., 2011*; *Grove et al., 1998*; *Kikkawa and Osumi, 2021*; *Quinn et al., 2009*; *Saulnier et al., 2013*). However, both *Dmrt* and *Gli3* are expressed broadly in the telencephalic medial neuroepithelium (in both the CH and beyond), and their overexpression was not reported to induce ectopic CH, suggesting that they play permissive rather than instructive roles in CH development (*Subramanian et al., 2009a*; *Subramanian and Tole, 2009b*). Other genes that regulate CH size, such as *Lhx2*, are expressed in the cortical neuroepithelium rather than CH itself. *Lhx2* confers cortical identity in the telencephalic neuroepithelium, cell-autonomously suppressing CH fate (*Mangale et al., 2008*; *Monuki et al., 2001*).

LIM-homeodomain transcription factor Lmx1a is one of the earliest markers of the CH. In contrast to the aforementioned genes, its expression in the telencephalon is limited to dorsal midline structures – the CH and non-neural choroid plexus epithelium (ChPe) (*Caronia-Brown et al., 2016*; *Chizhikov et al., 2019*; *Failli et al., 2002*). Previously, we found that loss of *Lmx1a* results in an aberrant contribution of the *Lmx1a*-lineage cells to the hippocampus, suggesting a function of this gene in establishing the CH/hippocampus boundary (*Chizhikov et al., 2010*). This phenotype, however, was traced to embryonic day (e) 10.5, when the dorsal midline has not yet fully differentiated into the ChPe and CH. Thus, it was unknown whether Lmx1a has a role in the CH development or its function is limited to the segregation of the dorsal midline lineage from the cortical neuroepithelium before the formation of the CH.

In this article, we investigated the role of *Lmx1a* in CH development by performing *Lmx1a* loss- and gain-of-function studies, complemented by RNAseq, in utero electroporation, and genetic experiments to identify downstream mediators of *Lmx1a* function in CH. We found that loss of *Lmx1a* compromises expression of numerous CH and CR cell markers, expression of Wnt signaling molecules, and cell cycle exit/differentiation and migration of CR cells, resulting in specific hippocampal abnormalities. Overexpression of *Lmx1a* was sufficient to induce key CH features in the cortical neuroepithelium, introducing *Lmx1a* as a master regulator of CH development.

## Results

### *Lmx1a*$^{-/-}$ mice exhibit a compromised DG associated with proliferation, glial scaffold, and migration abnormalities

Although *Lmx1a* is not expressed in the hippocampus (*Chizhikov et al., 2019*; *Chizhikov et al., 2010*), development of the hippocampus depends on CH (*Hevner, 2016*; *Subramanian et al., 2009a*) and, thus, can serve as a 'readout' of CH functioning in *Lmx1a*$^{-/-}$ mice. Extending previous histological studies (*Kuwamura et al., 2005*; *Sekiguchi et al., 1992*), we found a modest (~17%) reduction in the length of the hippocampal CA1-3 fields (*Figure 1A–C*), but a dramatic reduction of the HF that overlays the DG (~55% reduction, *Figure 1A, B and D*) and the number of DG (Prox1+) neurons (~78% reduction, *Figure 1E–G*) in postnatal day 21 (P21) *Lmx1a*$^{-/-}$ mice. Input resistance, a major

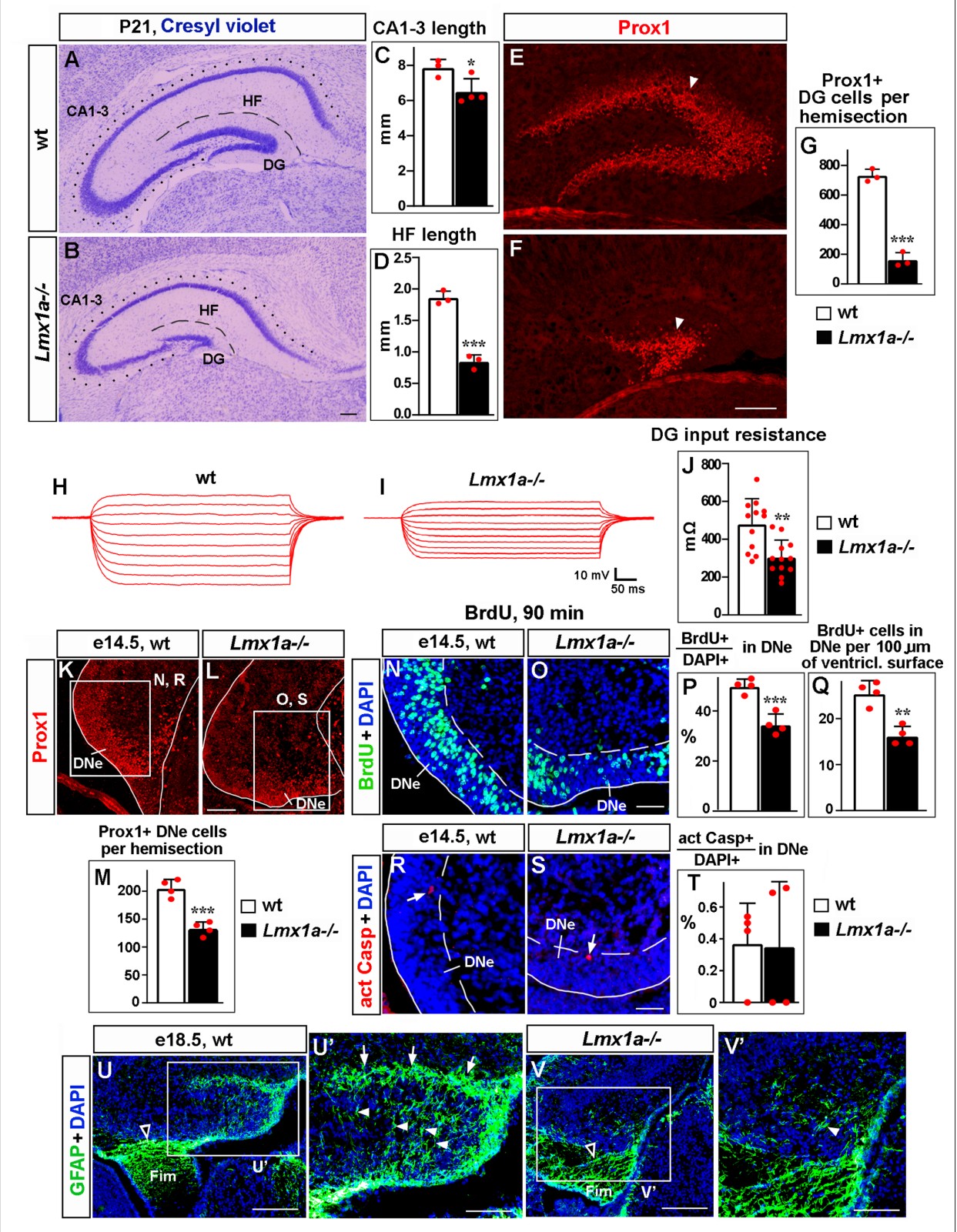

**Figure 1.** Loss of *Lmx1a* compromises dentate gyrus (DG) development. In this and other figures, panels show coronal sections immunostained with indicated antibodies (unless noted otherwise). High-magnification images show regions boxed in adjacent panels or diagrams. wt, wild-type controls. (**A–G**) In P21 *Lmx1a*[-/-] mice, length of the CA1-CA3 hippocampal fields (dotted line in **A, B**), length of the hippocampal fissure (HF) that overlays the DG (dashed line in **A, B**), and the number of Prox1+ DG neurons (arrowhead in **E, F**) were reduced (**C, D, G**). *p<0.05, ***p<0.001, two-tailed *t*-test, n = 3–4

*Figure 1 continued on next page*

*Figure 1 continued*

animals per genotype. (**H–J**) Current–voltage curves (**H, I**) and a bar graph (**J**) showing a reduced input resistance of DG granule neurons in P21 *Lmx1a*$^{-/-}$ mice. \*\*p<0.01, two-tailed *t*-test, N = 12 DG neurons from n = 5 mice per genotype. (**K–M**) Identification of the dentate neuroepithelium (DNe) by Prox1 immunostaining. Fewer (Prox1+) DNe cells were detected in *Lmx1a*$^{-/-}$ mutants at e14.5. \*\*\*p<0.001, two-tailed *t*-test, n = 4 embryos per genotype. (**N–Q**) Proliferation (% of BrdU+ cells among DAPI+ cells after a 90 min BrdU pulse [**P**] or the number of BrdU+ cells per 100 μm of the ventricular surface [**Q**]) was reduced in the DNe (identified by Prox1 immunostaining) of e14.5 *Lmx1a*$^{-/-}$ embryos. \*\*p<0.01, \*\*\*p<0.001, two-tailed *t*-test, n = 4 embryos per genotype. (**R–T**) Arrows indicate activated Caspase 3+ (apoptotic) cells. Apoptosis (% of apoptotic cells among DAPI+ cells) was not different in the DNe of e14.5 *Lmx1a*$^{-/-}$ and wild-type embryos. p>0.05, two-tailed *t*-test, n = 4 embryos per genotype. (**U–V'**) By late embryogenesis, cortical hem (CH) transforms into fimbria (Fim). The GFAP+ fimbrial scaffold (open arrowheads) was still present (**U, V**), but the transhilar scaffold was severely diminished in *Lmx1a*$^{-/-}$ mice (**U', V'**). Arrowheads and arrows indicate GFAP+ glial fibers that cross the hilus and enrich at the HF, respectively. Scale bars: 250 μm (**A, B, E, F, U, V**); 100 μm (**K, L, U', V'**); 60 μm (**N, O, R, S**).

The online version of this article includes the following source data and figure supplement(s) for figure 1:

**Source data 1.** Data points for *Figure 1C, D, G, J, M, P, Q and T*.

**Figure supplement 1.** Normal hippocampal patterning but a reduced number of dentate gyrus (DG) neurons in *Lmx1a*$^{-/-}$ mice at P3.

**Figure supplement 1—source data 1.** Data points for *Figure 1—figure supplement 1E*.

**Figure supplement 2.** Abnormal distribution of cells migrated from the dentate neuroepithelium (DNe) in *Lmx1a*$^{-/-}$ mice.

**Figure supplement 2—source data 1.** Data points for *Figure 1—figure supplement 2E, H and I*.

**Figure supplement 3.** Normal apoptosis in the developing hippocampus in *Lmx1a*$^{-/-}$ mice.

**Figure supplement 3—source data 1.** Data points for *Figure 1—figure supplement 3C, F, G, J and M*.

determinant of neuronal excitability (*Yang et al., 2021*), was aberrantly low in DG neurons in *Lmx1a*$^{-/-}$ mice (*Figure 1H–J*). In P3 *Lmx1a*$^{-/-}$ mutants, Ctip2 immunohistochemistry (*Roy et al., 2019*) revealed intact patterning of the hippocampal subfields (*Figure 1—figure supplement 1A and B*), but already reduced number of DG neurons (*Figure 1—figure supplement 1C–E*). Thus, to identify the mechanisms mediating DG abnormalities in *Lmx1a*$^{-/-}$ mutants, we analyzed embryonic stages, focusing on processes known to shape the DG.

During embryonic development, DG progenitors proliferate in the DNe in the ventricular zone. Immunohistochemistry with antibodies specific for Prox1, which at e14.5 serves as a marker for DNe (*Lavado et al., 2010*), revealed the presence of fewer DNe cells in *Lmx1a*$^{-/-}$ mutants than in their wild-type littermates (*Figure 1K–M*). A short (90 min) BrdU pulse, used to label cells in the S phase of the cell cycle, revealed a reduced proliferation (*Figure 1N–Q*), but immunohistochemistry with antibodies specific for activated Caspase 3 revealed no difference in the number of apoptotic cells in the DNe of *Lmx1a*$^{-/-}$ embryos relative to their wild-type littermates at this early stage (*Figure 1R–T*).

Around e14.5, progenitors begin exiting the DNe, forming the dentate migratory stream (DMS) and then migrating to the fimbria-dentate junction (FDJ) and to the DG (*Nelson et al., 2020*). To study cell migration, we performed BrdU labeling experiments (*Cai et al., 2018*). Progenitors in the DNe were labeled at e14.5, and the distribution of BrdU-labeled migrated cells was analyzed at e16 and e18.5 (*Figure 1—figure supplement 2A–H*). In e16 wild-type embryos, we observed a stream of BrdU+ cells that extended via the DMS to FDJ, but not to the DG, which is yet to form at this stage (*Figure 1—figure supplement 2C*). Compared to wild-type controls, in *Lmx1a* mutants, a larger fraction of migrated BrdU+ cells was located at the beginning of their migratory route (DMS), and fewer of them were found in the FDJ (*Figure 1—figure supplement 2C–E*). In control embryos, at e18.5, BrdU+ cells were found in the DMS, FDJ, and DG regions (*Figure 1—figure supplement 2F*). Similar to e16, at e18.5, *Lmx1a* mutants exhibited a larger fraction of migrated BrdU+ cells in the DMS, and smaller fractions of BrdU+ cells in the FDJ and DG regions (*Figure 1—figure supplement 2F–H*), suggesting a migration abnormality. The total number of migrated BrdU+ cells (i.e., those located in the DMS + FDJ + DG regions) was significantly lower in *Lmx1a*$^{-/-}$ embryos compared to wild-type controls (*Figure 1—figure supplement 2F, G and I*), which is consistent with a reduced number of Prox1+ DNe progenitors and reduced DNe proliferation in *Lmx1a*$^{-/-}$ mutants (*Figure 1K–Q*).

Fimbrial and transhilar glial scaffolds are necessary for the migration of progenitors from the DNe to DG (*Caramello et al., 2021*; *Hevner, 2016*; *Hodge et al., 2013*). While the fimbrial scaffold appeared grossly normal, the transhilar scaffold was diminished in e18.5 *Lmx1a* mutants (*Figure 1U–V'*). Analysis of apoptosis using anti-activated Caspase 3 immunohistochemistry at late embryonic (e16, e18.5) and postnatal (P3, P21) stages, when migration from the DNe occurs and/or the DG is forming, revealed

no difference between control and *Lmx1a⁻/⁻* mice (*Figure 1—figure supplement 3*). Taken together, our data indicate that proliferation and, likely, migration abnormalities, but not elevated apoptosis, contribute to DG deficits in *Lmx1a⁻/⁻* mice.

## Reduced exit of CH progenitors from the cell cycle and compromised migration of CH-derived CR cells in *Lmx1a⁻/⁻* mice

Next, we analyzed CH, where *Lmx1a* is expressed during embryogenesis. A short BrdU pulse and anti-activated Caspase 3 immunohistochemistry (*Ivaniutsin et al., 2009*; *Miquelajauregui et al., 2007*) revealed normal proliferation and apoptosis in the CH of *Lmx1a* mutants (*Figure 2—figure supplement 1*). In contrast, BrdU/Ki67 immunohistochemistry after a 24 hr BrdU pulse (*Chizhikov et al., 2019*) revealed a smaller fraction of progenitors exiting the cell cycle in *Lmx1a⁻/⁻* embryos (*Figure 2A–C*), suggesting that *Lmx1a* is required for normal differentiation of CH progenitors.

Since CH progenitors differentiate into CR cells (*Dixit et al., 2011*; *Gu et al., 2011*), we studied CR cells. After exiting the CH, CR cells first migrate tangentially along the FDJ and then radially into the hippocampal field to form the HF (*Causeret et al., 2021*). Immunohistochemistry against the CR cell marker Reelin (*Hodge et al., 2013*; *Ledonne et al., 2016*; *Siegenthaler and Miller, 2008*) revealed that in e15.5 wild-type embryos many CR cells have populated the HF and few remained at the FDJ. In contrast, in *Lmx1a⁻/⁻* littermates, many CR cells were still at the FDJ, and the HF was abnormally short (*Figure 2D–F*). CR cell distribution and HF abnormalities persisted at P3 (*Figure 2—figure supplement 2*), indicating that loss of *Lmx1a* leads to severe long-lasting deficits in the migration of CR cells.

## Genes and developmental processes misregulated in the CH of *Lmx1a⁻/⁻* mice based on RNAseq analysis

To identify CH genes regulated by *Lmx1a*, we performed RNAseq analysis. For these experiments, we isolated CH by laser capture microdissection (LCM) from wild-type and *Lmx1a⁻/⁻* embryos at e13.5, when the hippocampus begins to develop, and the CR cells that populate the HF emerge from the CH (*Gu et al., 2011*; *Nelson et al., 2020*). We found that 612 genes were significantly downregulated, and 457 genes were significantly upregulated in the CH of *Lmx1a* mutants compared to wild-type littermates (*Figure 2G and H*). Notably, these genes included *Cux2* (*Figure 2H*, red dot), the only previously known *Lmx1a* downstream gene in the CH (*Fregoso et al., 2019*), validating our experimental approach.

Of the 100 genes most misregulated in *Lmx1a⁻/⁻* CH (false discovery rate [FDR] < 0.05), 55 had in situ hybridization images in public databases (Gene Paint, Eurexpress, Allen Brain Atlas). Interestingly, 14 of these genes (~25%) were specifically or predominantly expressed in the CH and were downregulated in *Lmx1a* mutants (*Figure 2H*, blue dots, and *Figure 2—figure supplement 3*), including a previously described CH marker *Lmo2* (*Mangale et al., 2008*) and 13 novel CH markers (*Slc17a8*, *Ccdc3*, *Adamts1*, *Galnt14*, *Camk2a*, *Arfgef3*, *Rdh10*, *Plxnc1*, *Kdr*, *Asb4*, *Slc14a2*, *Peg3*, and *Slit3*). Functionally, these genes varied considerably, ranging from transcription factors (*Lmo2*) to transporters of molecules (*Slc17a8*, *Slc14a2*) to lipid and carbohydrate metabolism regulators (*Ccdc3*, *Rhd10*) to transmembrane receptors (*Plxnc1*).

Having observed CR cell abnormalities in *Lmx1a* mutants (*Figure 2D–E'*), we studied whether our newly identified Lmx1a downstream targets include genes associated with CR cells, taking advantage of recently identified datasets of genes enriched in CR cells (*Franzén et al., 2019*; *Li et al., 2021*). Eleven known CR cell-enriched genes were downregulated (*Car10*, *Lhx1*, *Trp73*, *Cntnap2*, *Reln*, *Cdkn1a*, *Chst8*, *Spock1*, *Akap6*, *Cacna2d2*, and *Tbr2*) and six were upregulated (*Tmem163*, *Sema6a*, *Lhx5*, *Lingo1*, *Edil3*, and *Mapk8*) in the CH of *Lmx1a⁻/⁻* embryos (green dots in *Figure 2H*). Thus, loss of *Lmx1a* diminishes the expression of a diverse set of CH markers and compromises the expression of CR cell-associated genes in the CH.

To further test for an intrinsic role of *Lmx1a* in the CH, we in utero electroporated the CH of e11 wild-type embryos with either previously validated anti-*Lmx1a* shRNA (*Fregoso et al., 2019*) or control shRNA together with GFP. At e13.5, GFP+ cells in the CH were isolated by LCM, and gene expression was analyzed by qRT-PCR (*Figure 2—figure supplement 4*). In these experiments, we confirmed the downregulation of all seven tested Lmx1a downstream targets, identified by RNAseq, including both CH and CR markers/key developmental genes (see below) (*Figure 2—figure supplement 4E–K*), supporting an intrinsic function of *Lmx1a* in the CH.

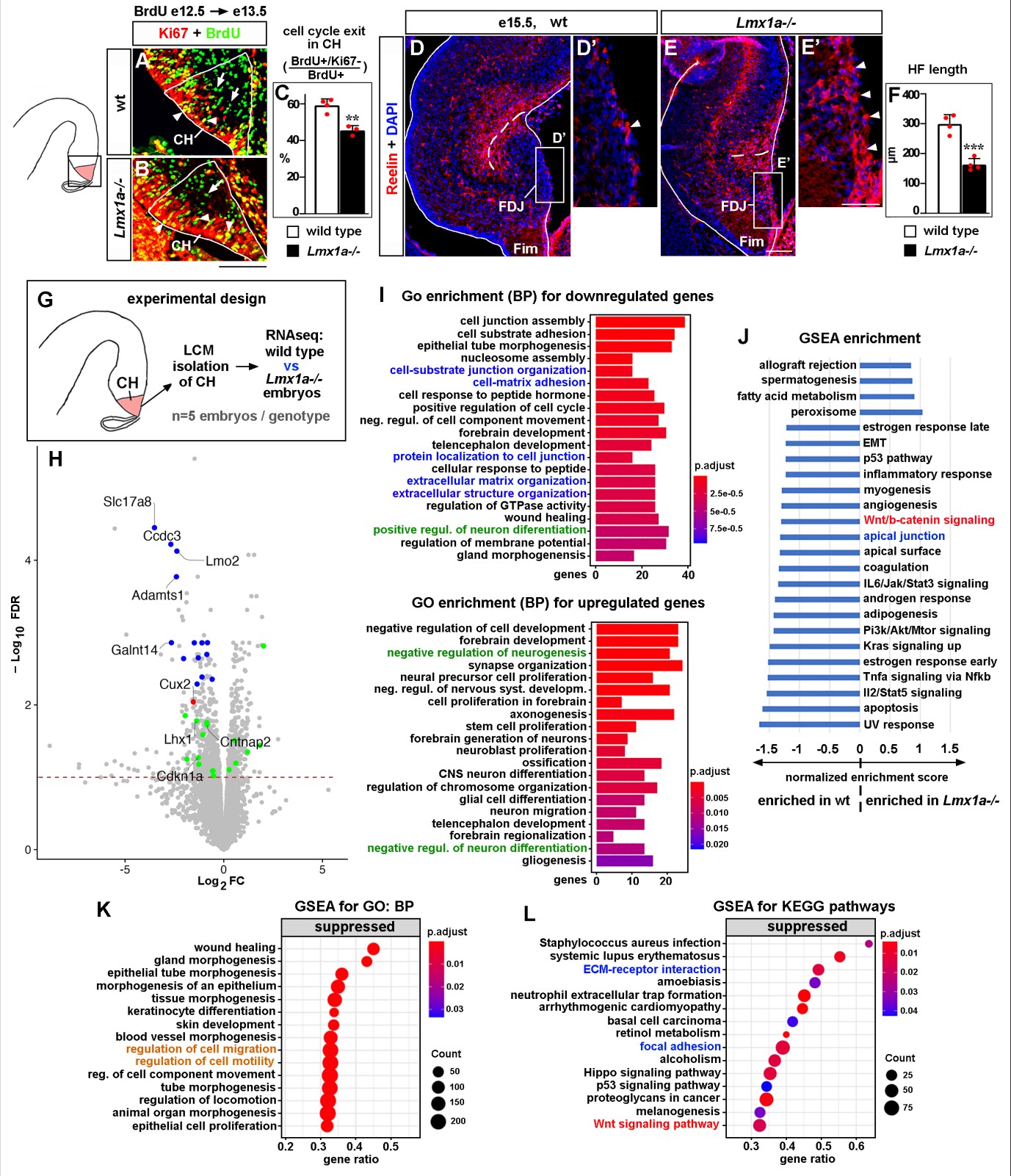

**Figure 2.** Developmental processes compromised in the cortical hem (CH) of *Lmx1a*⁻/⁻ mice. (**A–C**) A reduced fraction (%) of progenitors exited the cell cycle in the CH of *Lmx1a*⁻/⁻ embryos. 24 hr after BrdU injection, progenitors that exited the cell cycle were BrdU+/Ki67- (green, arrows); progenitors that re-entered the cell cycle were BrdU+/Ki67+ (yellow, arrowheads). **p<0.01, two-tailed *t*-test, n = 3–4 embryos per genotype. (**D–F**) By e15.5, in wild-type embryos, many Cajal–Retzius (CR) cells have already migrated into the hippocampal fissure (HF) area, and few CR cells remain at the fimbria-dentate

*Figure 2 continued on next page*

*Figure 2 continued*

junction (FDJ) surface (**D, D'**, arrowhead). In contrast, many CR cells were still located at the FDJ surface in *Lmx1a*$^{-/-}$ littermates (**E, E'**, arrowheads), which was associated with a reduced HF length (dashed line in **D–F**). ***p<0.001, two-tailed *t*-test, n = 4 embryos per genotype. (**G**) Experimental design for RNAseq analysis. (**H**) Volcano plot of transcripts detected by RNAseq. Transcripts above the dashed line (false discovery rate [FDR] = 0.1) were considered differentially expressed in *Lmx1a*$^{-/-}$ CH. CH markers, identified among the 100 most misregulated genes, are shown in blue (also see *Figure 2—figure supplement 3*); genes associated with CR cells (*Franzén et al., 2019*; *Li et al., 2021*) are shown in green; the only previously identified *Lmx1a* target in the CH, *Cux2* (*Fregoso et al., 2019*), is shown in red. (**I**) GO (biological process, BP) enrichment analysis for the genes differentially expressed in *Lmx1a*$^{-/-}$ CH. (**J**) Pathways enriched in wild-type and *Lmx1a*$^{-/-}$ CH based on Gene Set Enrichment Analysis (GSEA) of RNAseq data. (**K, L**) Top 15 pathways/processes suppressed in the *Lmx1a*$^{-/-}$ CH based on GSEA for GO (BP) and GSEA for Kyoto Encyclopedia of Genes and Genomes (KEGG) analyses. Processes/pathways related to neuronal differentiation, cell migration, Wnt signaling, and tissue integrity are highlighted in green, orange, red, and blue, respectively. Scale bars: 100 μm (**A, B, D, E**); 50 μm (**D', E'**).

The online version of this article includes the following source data and figure supplement(s) for figure 2:

**Source data 1.** Data points for *Figure 2C and F*.

**Figure supplement 1.** Normal proliferation and apoptosis in the cortical hem (CH) of *Lmx1a*$^{-/-}$ embryos at e12.5.

**Figure supplement 1—source data 1.** Data points for *Figure 2—figure supplement 1C and F*.

**Figure supplement 2.** Abnormal distribution of Cajal–Retzius (CR) cells in P3 *Lmx1a*$^{-/-}$ mice.

**Figure supplement 2—source data 1.** Data points for *Figure 2—figure supplement 2C*.

**Figure supplement 3.** Reduced expression of cortical hem (CH) markers in *Lmx1a*$^{-/-}$ embryos.

**Figure supplement 4.** *Lmx1a* downregulation specifically in the cortical hem (CH) reduces the expression of CH markers and key CH developmental genes.

**Figure supplement 4—source data 1.** Data points for *Figure 2—figure supplement 4E–K*.

To gain further insight into the developmental processes and pathways affected in the CH by Lmx1a loss, we performed bioinformatics analysis of our RNAseq data (*Figure 2I–L*). Interestingly and consistently with a reduced cell cycle exit of CH progenitors (*Figure 2A–C*), Gene Ontology (GO) enrichment analysis revealed that the 'positive regulation of neuronal differentiation' category was overrepresented in genes significantly downregulated, while the 'negative regulation of neurogenesis' and 'negative regulation of neuronal differentiation' categories were overrepresented in genes significantly upregulated in the CH of *Lmx1a*$^{-/-}$ embryos (*Figure 2I*, highlighted in green). Consistent with the abnormal distribution of CR cells (*Figure 2D–E'*), Gene Set Enrichment Analysis (GSEA) for GO revealed that 'regulation of cell migration' and 'regulation of cell motility' processes were suppressed in *Lmx1a*$^{-/-}$ mutants (*Figure 2K*, highlighted in orange). In addition, the GSEA revealed enrichment of Wnt/β-catenin signaling in wild-type relative to *Lmx1a*$^{-/-}$ CH (*Figure 2J*). Similarly, GSEA for Kyoto Encyclopedia of Genes and Genomes (KEGG) revealed suppressed Wnt signaling in the CH of *Lmx1a* mutant mice (*Figure 2L*, highlighted in red). Finally, GO enrichment, GSEA, and GSEA for KEGG analyses predicted compromised cell–cell and cell–extracellular matrix adhesion in *Lmx1a*$^{-/-}$ CH (*Figure 2I, J and L*, related processes are highlighted in blue), suggesting that abnormal cell adhesion contributes to previously reported aberrant dispersion of *Lmx1a*-lineage cells into the adjacent hippocampus in *Lmx1a* mutants (*Chizhikov et al., 2010*). Thus, *Lmx1a* is required for normal expression of diverse CH markers and CR cell genes, and our bioinformatics analysis supports the role of *Lmx1a* in the differentiation and migration of CR cells, and canonical Wnt signaling.

### *Lmx1a* regulates expression of *Wnt3a* to promote DG development

Canonical Wnt signaling promotes proliferation in the DNe and transhilar scaffold formation (*Li and Pleasure, 2005*; *Zhou et al., 2004*; *Parichha et al., 2022*), suggesting a hypothesis that *Lmx1a* regulates DG morphogenesis at least partially via secreted Wnts produced in the CH. Our RNAseq analysis revealed that two Wnts known to activate canonical Wnt signaling, Wnt2b and Wnt9a, both expressed specifically in the CH, were significantly (FDR < 0.05) downregulated in *Lmx1a*$^{-/-}$ mutants (*Figure 3—figure supplement 1*). *Wnt2b* knockout mice have a normally sized DG (*Tsukiyama and Yamaguchi, 2012*), while Wnt9a, although capable of activating canonical Wnt signaling in the liver (*Matsumoto et al., 2008*), primarily acts as non-canonical Wnt ligand (*Nie et al., 2020*). In contrast, the knockout of CH-specific *Wnt3a* resulted in a nearly absent DG (*Lee et al., 2000*). Our RNAseq analysis showed that *Wnt3a* was downregulated in the CH of e13.5 *Lmx1a*$^{-/-}$ mice, although the difference did not reach a statistical significance (FDR = 0.13, *Figure 3—figure supplement 1*). However, quantification of in

situ hybridization signal revealed a significant (p<0.01) downregulation of *Wnt3a* expression in the CH of *Lmx1a*$^{-/-}$ embryos at e14 (*Figure 3A–C*), just before we detected a reduced proliferation in the DNe (*Figure 1N–Q*). qRT-PCR analysis of LCM-isolated CH confirmed downregulation of *Wnt3a* in *Lmx1a*$^{-/-}$ mutants at this stage (*Figure 3D and E*). While *Wnt3a* expression was normal in the CH of *Lmx1a*$^{+/-}$ or *Wnt3a*$^{+/-}$ embryos, loss of one copy of *Lmx1a* on the *Wnt3a*$^{+/-}$ background (in *Lmx1a*$^{+/-}$;*Wnt3a*$^{+/-}$ double heterozygous embryos) resulted in a reduced expression of *Wnt3a* in the CH, providing additional evidence for regulation of *Wnt3a* expression by *Lmx1a* (*Figure 3E*). Knockdown of *Lmx1a* in the CH resulted in a reduced expression of *Wnt3a* (*Figure 2—figure supplement 4I*), supporting an intrinsic role for *Lmx1a* in regulating *Wnt3a* expression.

One approach to test whether *Lmx1a* regulates DG development via *Wnt3a* could be *Wnt3a* in utero electroporation rescue experiments in *Lmx1a*$^{-/-}$ mice. However, this approach is likely to produce higher levels of Wnt3a in at least some cells of *Lmx1a*$^{-/-}$ embryos relative to endogenous Wnt3a levels in wild-type mice. Normal DG development is a complex process that is particularly sensitive to the level and dynamics of canonical Wnt signaling (*Arredondo et al., 2020*; *Zhou et al., 2004*; *Parichha et al., 2022*). Thus, overactivation of the Wnt signaling pathway may cause additional abnormalities, thereby complicating the interpretation of results from *Wnt3a* electroporation experiments aimed to rescue *Lmx1a*$^{-/-}$ hippocampal phenotypes. Instead, we used the alternative approach of testing whether *Lmx1a* and *Wnt3a* co-regulate hippocampal development by analyzing *Lmx1a/Wnt3a* double heterozygotes rather than *Wnt3a* overexpression rescue experiments in *Lmx1a*$^{-/-}$ mice. In contrast to wild-type or single-gene heterozygous (*Lmx1a*$^{+/-}$ or *Wnt3a*$^{+/-}$) mice, their double heterozygous *Lmx1a*$^{+/-}$;*Wnt3a*$^{+/-}$ littermates exhibited a reduced number (*Figure 3F–J*) and input resistance (*Figure 3K and L*) of DG neurons, and transhilar glial scaffold abnormalities (*Figure 3M–O*). A reduced number of DG neurons was associated with reduced proliferation in DNe (*Figure 3P–R*). Although these double heterozygote experiments alone do not necessarily show that one gene acts via the other, as two genes may act via parallel pathways, reduced expression of *Wnt3a* in *Lmx1a*$^{-/-}$ embryos and down-regulation of *Wnt3a* expression in *Lmx1a*$^{+/-}$;*Wnt3a*$^{+/-}$ embryos relative to *Wnt3a*$^{+/-}$ embryos show that *Lmx1a* acts upstream of *Wnt3a*, thus, suggesting that *Lmx1a* promotes DG development, at least partially, by modulating expression of *Wnt3a*.

## *Lmx1a* regulates expression of *Tbr2* to promote the migration of CR cells and HF and transhilar scaffold formation

CR cells promote HF and transhilar glial scaffold formation (*Frotscher et al., 2003*; *Hodge et al., 2013*; *Meyer et al., 2004*; *Meyer et al., 2019*). Having observed CR cell migration deficits in *Lmx1a*$^{-/-}$ mutants (*Figure 2D–F*), we studied *Lmx1a*-dependent mechanisms of CR cell migration. For this purpose, we searched the literature to find whether mutants for CR cell-associated genes misregulated in *Lmx1a*$^{-/-}$ CH (*Figure 2H*, green dots) have phenotypes similar to those in *Lmx1a*$^{-/-}$ mutants. Interestingly, *Nestin-Cre;Tbr2*$^{floxed/floxed\ (F/F)}$ mice exhibited CR cell migration, HF, and glial scaffold abnormalities (*Hodge et al., 2013*) that resemble those in *Lmx1a*$^{-/-}$ mice. While *Tbr2* is expressed in differentiating CR cells, this gene is also expressed in intermediate hippocampal (DG) progenitors. *Nestin-Cre* mice delete floxed sequences in both the CH (a site of origin of CR cells) and hippocampal progenitors (*Hodge et al., 2013*), making it difficult to identify the developmental origin of the CR/hippocampal phenotypes in *Nestin-Cre;Tbr2*$^{F/F}$ mice.

To study the intrinsic role of *Tbr2* in CR cells arising from the CH, we generated and analyzed *Lmx1a-Cre;Tbr2*$^{F/F}$ mice, in which *Tbr2* was specifically inactivated in the CH but still expressed in hippocampal progenitors (*Figure 4—figure supplement 1A–C'*). Similar to *Nestin-Cre;Tbr2*$^{F/F}$ mice (*Hodge et al., 2013*), *Lmx1a-Cre;Tbr2*$^{F/F}$ mice exhibited aberrant accumulation of Reelin+ CR cells at the FDJ, associated with a reduced HF length (*Figure 4—figure supplement 1D–F*), supporting the conclusion that *Tbr2* acts intrinsically in the CR cell lineage to regulate migration of CR cells and HF/DG morphogenesis.

CR cells arise from progenitors in the ventricular layer of the CH. Concurrent with differentiation, they migrate radially to the CH marginal zone, where CR cells accumulate before initiating tangential migration toward the hippocampal fissure (*Bielle et al., 2005*; *Causeret et al., 2021*; *Hodge et al., 2013*; *Meyer et al., 2004*). Co-immunohistochemistry revealed that in e13 wild-type embryos virtually all Tbr2+ cells located in the marginal zone of CH express a CR marker p73 and, thus, were specified CR cells. In contrast, many Tbr2+ cells in the ventricular and intermediate zones of the CH

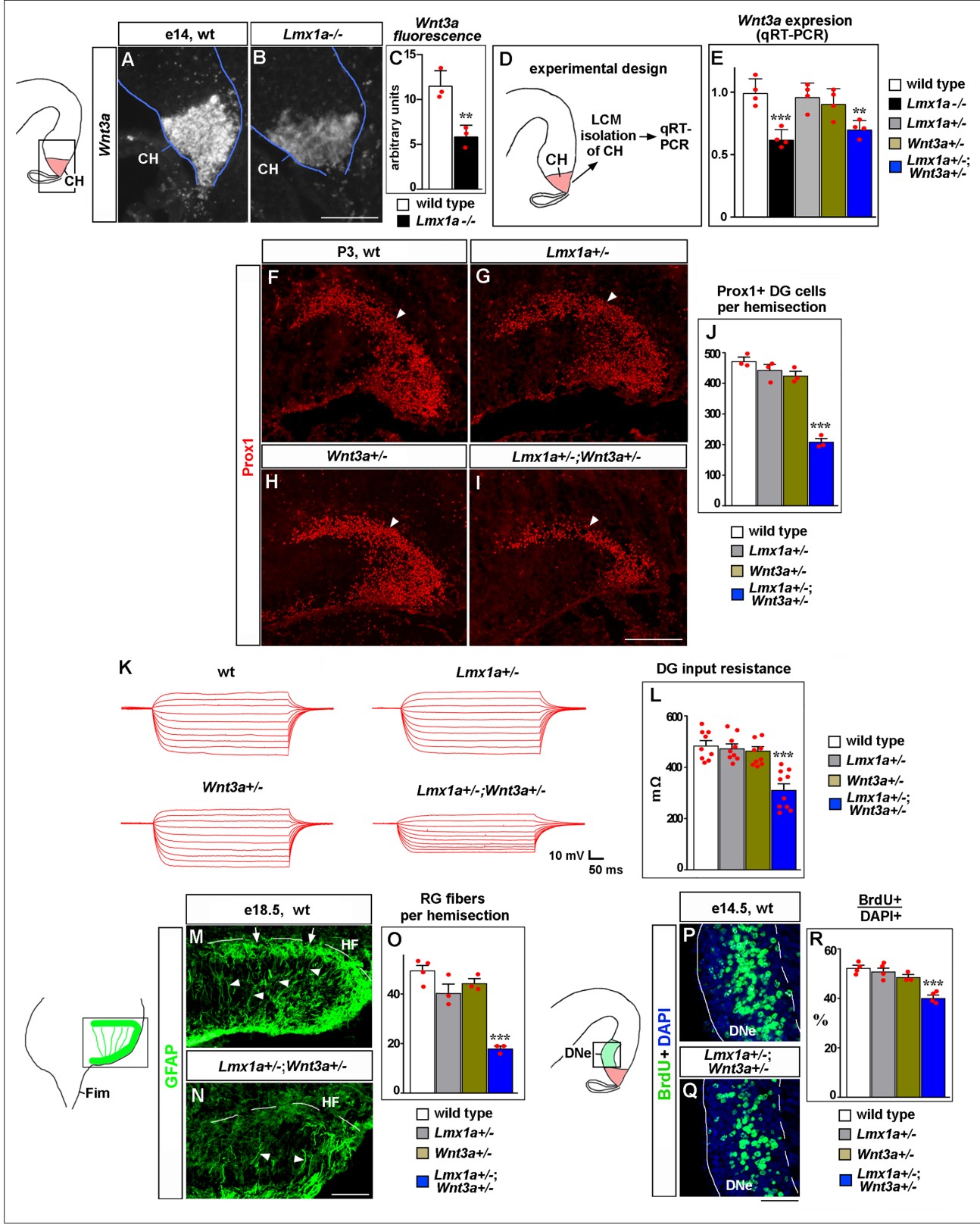

**Figure 3.** *Lmx1a* regulates dentate gyrus (DG) development by modulating expression of *Wnt3a*. (**A–C**) The intensity of *Wnt3a* in situ hybridization signal (white) in the cortical hem (CH) was reduced in e14 *Lmx1a⁻/⁻* embryos. **p<0.01, two-tailed *t*-test, n = 3 embryos per genotype. (**D, E**) Analysis of *Wnt3a* expression in the CH by qRT-PCR. Experimental design (**D**): CH of e14 embryos of different genotypes was isolated by laser capture microdissection (LCM) and analyzed by qRT-PCR (**E**). *Wnt3a* expression was significantly reduced in the CH of *Lmx1a⁻/⁻* and *Lmx1a⁺/⁻;Wnt3a⁺/⁻* double

*Figure 3 continued on next page*

*Figure 3 continued*

heterozygotes, but not *Lmx1a*$^{+/-}$ or *Wnt3a*$^{-/-}$ single heterozygote embryos compared to wild-type controls. ***p<0.001, **p<0.01, one-way ANOVA with Tukey's post hoc test, n = 4 embryos per genotype. (F–J) The number of Prox1+ DG neurons (arrowhead) was reduced in *Lmx1a*$^{+/-}$;*Wnt3a*$^{+/-}$ double heterozygotes, but not single-gene heterozygotes, compared to wild-type controls at P3. ***p<0.001, one-way ANOVA with Tukey's post hoc test, n = 3 mice per genotype. (K, L) Current–voltage curves (K) and a bar graph (L) showing a reduced input resistance of DG granule neurons in *Lmx1a*$^{+/-}$;*Wnt3a*$^{+/-}$ double heterozygotes, but not single-gene heterozygotes, compared to wild-type controls at P21. ***p<0.001, one-way ANOVA with Tukey's post hoc test, N = 9–10 DG neurons from n = 4–5 mice per genotype. (M–O) The number of GFAP+ transhilar glial fibers was reduced in *Lmx1a*$^{+/-}$;*Wnt3a*$^{+/-}$ double heterozygotes, but not single-gene heterozygotes, compared to wild-type controls. ***p<0.001, one-way ANOVA with Tukey's post hoc test, n = 3–4 mice per genotype. Arrowheads and arrows indicate fibers that cross the hilus and enrich at the hippocampal fissure (HF), respectively. (P–R) Proliferation (% of BrdU+ cells after a 90 min BrdU pulse) in dentate neuroepithelium (DNe) was reduced in *Lmx1a*$^{+/-}$;*Wnt3a*$^{+/-}$ double heterozygotes, but not single-gene heterozygotes, compared to wild-type controls. ***p<0.001, one-way ANOVA with Tukey's post hoc test, n = 3–4 mice per genotype. Scale bars: 100 μm (A, B, M, N); 200 μm (F–I); 50 μm (P, Q).

The online version of this article includes the following source data and figure supplement(s) for figure 3:

**Source data 1.** Data points for *Figure 3C, E, J, L, O and R*.

**Figure supplement 1.** Canonical Wnts misregulated in the cortical hem (CH) of *Lmx1a*$^{-/-}$ embryos based on RNAseq analysis.

were p73-negative, presumably representing less differentiated cells (*Figure 4—figure supplement 2*). To get a sense of the Tbr2 expression profile in *Lmx1a*$^{-/-}$ mutants, we divided the CH into three equally-sized bins along the radial axis, namely the ventricular, intermediate, and marginal zones, at e13, when the CR cells that predominantly populate the HF emerge from the CH (*Gu et al., 2011*). Consistent with our RNAseq data (*Figure 4A*), immunohistochemistry revealed that loss of *Lmx1a* reduced Tbr2 expression in the e13 CH across all three bins, but most significantly in the marginal zone (p<0.001, *Figure 4B and C*). In contrast, expression of Tbr2 in the hippocampal progenitors was not affected (*Figure 4B and D*). Knockdown of *Lmx1a* in the CH also resulted in reduced expression of *Tbr2* (*Figure 2—figure supplement 4J*), supporting an intrinsic role for *Lmx1a* in the regulation of *Tbr2* expression.

Since CR cell migration and DG morphogenesis are complex processes that require precise expression levels of key genes (*Gil et al., 2014*; *Ha et al., 2020*; *Hevner, 2016*), to study whether *Tbr2* and *Lmx1a* co-regulate CR cells and DG development, we analyzed *Lmx1a/Tbr2* double heterozygotes rather than performing *Tbr2* overexpression rescue experiments in *Lmx1a*$^{-/-}$ mice. In contrast to wild-type controls or single-gene heterozygotes (*Lmx1a*$^{+/-}$ or *Lmx1a-Cre;Tbr2*$^{+/F}$, in which one copy of *Tbr2* was specifically deleted in CH), their *Lmx1a*$^{+/-}$;*Lmx1a-Cre;Tbr2*$^{+/F}$ double-gene heterozygous littermates revealed aberrant accumulation of Reelin+ CR cells at the FDJ, reduced HF length and transhilar scaffold abnormalities (*Figure 4E–F* and *Figure 4—figure supplement 3*). These phenotypes were not associated with a delayed exit of CH progenitors from the cell cycle (*Figure 4G and H*). Taken together with reduced expression of Tbr2 in the *Lmx1a*$^{-/-}$ CH, these data suggest that *Lmx1a* modulates *Tbr2* expression to promote migration of CR cells and HF/transhilar glial scaffold formation.

## *Lmx1a* promotes cell cycle exit and differentiation of CR cells in the CH via *Cdkn1a*

One of the CR markers misregulated in the *Lmx1a*$^{-/-}$ CH (*Figure 2H*) was *Cdkn1a*, the gene known to promote exit from the cell cycle in several non-neural and neural cell types (*Siegenthaler and Miller, 2005*; *Xiao et al., 2020*). In the CH, expression of *Cdkn1a* coincides with differentiation of CR cells (*Siegenthaler and Miller, 2008*). Consistent with our RNAseq analysis (*Figure 5A*), immunohistochemistry revealed reduced *Cdkn1a* expression in the CH of *Lmx1a*$^{-/-}$ mice at e13 (*Figure 5B–D*). Knockdown of *Lmx1a* in the CH resulted in a reduced expression of *Cdkn1a* (*Figure 2—figure supplement 4K*), supporting an intrinsic role for *Lmx1a* in regulating *Cdkn1a* expression.

To study whether decreased *Cdkn1a* expression mediates a reduced cell cycle exit of CH progenitors in *Lmx1a*$^{-/-}$ embryos (*Figure 2A–C*), we used immunohistochemistry with antibodies specific for Ki67, which labels cycling progenitors. As the presence/absence of Ki67 expression is a simpler output than complex DG morphogenesis and long-range migration of CR cells, we performed *Cdkn1a* overexpression rescue studies using in utero electroporation of the CH at e11. The ventricular layer of the CH that borders the lateral ventricles consists of progenitor cells, so it is expected that plasmids injected into the lateral ventricles and electroporated into the CH will target such progenitors. However, since electroporation can also target differentiated cells (*Govindan et al., 2018*), we first

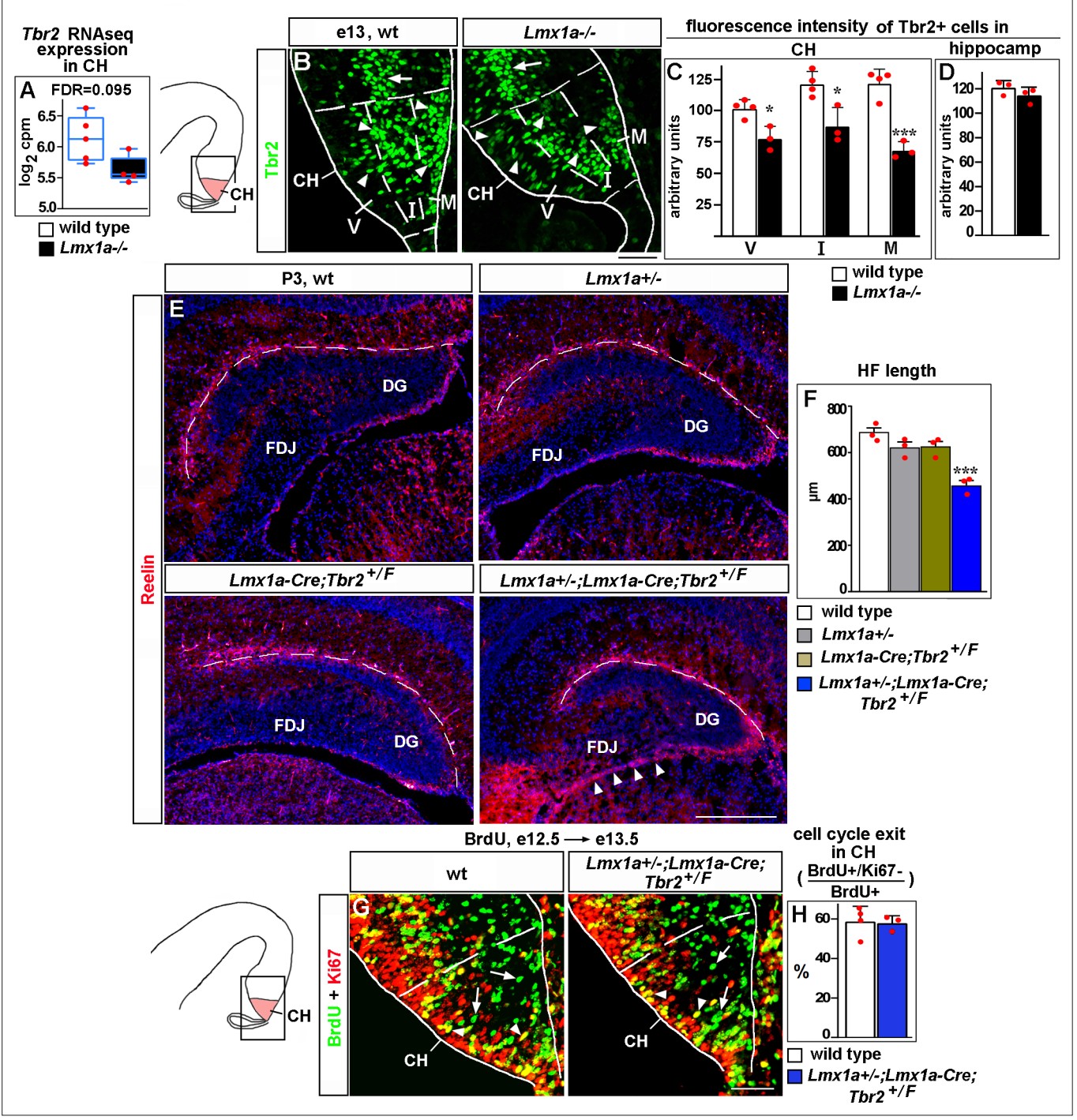

**Figure 4.** *Lmx1a* regulates Cajal–Retzius (CR) cell migration and hippocampal fissure (HF) formation by modulating expression of *Tbr2*. (**A**) Normalized read counts for *Tbr2* from the RNAseq experiment (*Figure 2G and H*). (**B–D**) Arrowheads indicate Tbr2+ cells in the CH (**B**). For quantification of intensity of Tbr2 immunostaining, the CH was divided into three equally sized bins along the radial axis: ventricular zone (V), intermediate zone (I), and marginal zone (M). In *Lmx1a⁻/⁻* mutants, Tbr2 staining intensity was reduced in all three zones of the CH, but most significantly in the marginal zone (**B, C**). In contrast, Tbr2+ cells in the hippocampal primordium (hippocampal intermediate progenitors, *Hodge et al., 2013*) (arrow in **B**) had similar Tbr2 immunofluorescence intensity in control and *Lmx1a⁻/⁻* embryos (**B, D**), suggesting that *Lmx1a* loss reduces *Tbr2* expression specifically in cells arising in the CH. (Dashed lines demarcate the CH boundaries, identified by Lmx1a immunostaining of adjacent sections, as described in the 'Materials and methods.') ***p<0.001, *p<0.05, two-tailed *t*-test, n = 3–4 embryos per genotype. (**E, F**) Reduced HF length (dashed line) and aberrant accumulation of CR cells at the fimbria-dentate junction (FDJ) surface (arrowheads) in *Lmx1a/Tbr2* double heterozygotes (*Lmx1a⁺/⁻;Lmx1a-Cre;Tbr2⁺/F* mice – *Lmx1a⁺/⁻* mice in which one copy of *Tbr2* was deleted specifically in the CH), but not single-gene heterozygotes, compared to wild-type controls at P3.

*Figure 4 continued on next page*

*Figure 4 continued*

***p<0.001, one-way ANOVA with Tukey's post hoc test, n = 3 mice per genotype. (**G, H**) Normal cell cycle exit of progenitors in the CH of *Lmx1a/Tbr2* double heterozygous embryos. 24 hr after BrdU injection, progenitors that exited the cell cycle were BrdU+/Ki67- (green, arrows); progenitors that re-entered the cell cycle were BrdU+/Ki67+ (yellow, arrowheads). p>0.05, two-tailed *t*-test, n = 3–4 mice per genotype. Scale bars: 50 μm (**B, G**); 200 μm (**E**).

The online version of this article includes the following source data and figure supplement(s) for figure 4:

**Source data 1.** Data points for *Figure 4C, D, F and H*.

**Figure supplement 1.** Loss of *Tbr2* in the cortical hem (CH) compromises migration of Cajal–Retzius (CR) cells and hippocampal fissure (HF) formation.

**Figure supplement 1—source data 1.** Data points for *Figure 4—figure supplement 1F*.

**Figure supplement 2.** Analysis of Tbr2 and p73 co-expression in the cortical hem (CH).

**Figure supplement 3.** Transhilar glial scaffold abnormalities in *Lmx1a/Tbr2* double heterozygous mice.

**Figure supplement 3—source data 1.** Data points for *Figure 4—figure supplement 3E*.

---

injected a GFP-encoding plasmid into the lateral ventricles, electroporated it in utero into the CH of e11 embryos and analyzed GFP+ cells after a short (15 hr) time period. This analysis revealed that virtually all (~95%) GFP+ cells were Ki67+ (progenitors) in both wild-type and *Lmx1a* mutant embryos (*Figure 5—figure supplement 1*), confirming that this system is appropriate to target progenitors.

Having established the assay, we electroporated the CH of e11 embryos with plasmids encoding either *Cdkn1a+GFP* or *GFP* alone (control) and analyzed embryos at a later (e13) stage to allow time for cell cycle exit/differentiation of electroporated cells. Expression of exogenous *Cdkn1a* in the CH of *Lmx1a$^{-/-}$* embryos increased the fraction of progenitors that exited the cell cycle (GFP+/Ki67- cells among GFP+ cells), making it comparable to that in controls (CH of wild-type embryos electroporated with *GFP* alone) (*Figure 5E–H*). Interestingly, *Cdkn1a* electroporation also normalized the number of differentiating (p73+) (*Meyer et al., 2004*; *Siegenthaler and Miller, 2008*) CR cells in *Lmx1a$^{-/-}$* embryos (*Figure 5I–L*), indicating that decreased *Cdkn1a* expression contributes to both reduced cell cycle exit in CH and decreased differentiation of CR cells in *Lmx1a$^{-/-}$* embryos.

## *Lmx1a* is sufficient to induce ectopic CH in the medial telencephalon

Having identified the role of *Lmx1a* in multiple aspects of CH development, we tested whether its expression is sufficient to confer CH fate in the telencephalon. We in utero electroporated *Lmx1a* into the medial telencephalon of wild-type embryos at e10.75, after the cortical neuroepithelium had already been specified by *Lhx2* (*Mangale et al., 2008*), and analyzed three key cortical hem parameters: (1) expression of the CH markers *Wnt3a* (which is also a major CH signaling molecule) and *Ccdc3*, (2) the appearance of ectopic p73+ CR cells (*Meyer et al., 2004*; *Siegenthaler and Miller, 2008*), and (3) the suppression of *Lhx2* expression, a key feature of native CH (*Mangale et al., 2008*).

In contrast to *GFP* controls, *Lmx1a+GFP* electroporation into medial telencephalon induced ectopic *Wnt3a* and *Ccdc3*. Extensive overlap of GFP expression with *Wnt3a* and *Ccdc3* signals suggested cell-autonomous induction of these genes by *Lmx1a* in these experiments (*Figure 6A–D'* and *Figure 6—figure supplement 1A–D'*). Normally, CR cells, identified by the expression of their specific marker p73, arise deep in the CH and then migrate along the outer surface of the telencephalic neuroepithelium (*Meyer et al., 2004*; *Siegenthaler and Miller, 2008*). In contrast to *GFP* controls, in *Lmx1a*-electroporated samples, we observed numerous ectopic p73+ cells deep in the neuroepithelium beyond the CH (*Figure 6E–H'*), which was associated with a dramatic increase in the total number of p73+ cells (*Figure 6E–I*). Finally, quantification of the immunohistochemistry signal revealed a significant reduction of Lhx2 expression in hippocampal primordium cells electroporated with *Lmx1a+GFP* versus those electroporated with *GFP* alone (*Figure 6J–L*). Notably, electroporation of *Lmx1a* did not induce expression of the choroid plexus marker Ttr (*Figure 6—figure supplement 1E–F'*). Interestingly, when electroporated into the lateral cortex, *Lmx1a* did not induce expression of *Wnt3a* (*Figure 6—figure supplement 2*). Thus, *Lmx1a* expression specifically induces key CH features in the medial telencephalic neuroepithelium.

## Discussion

While the critical role of signaling centers in the development of the vertebrate CNS has been recognized for decades, very little is known about how distinct cellular and molecular mechanisms are

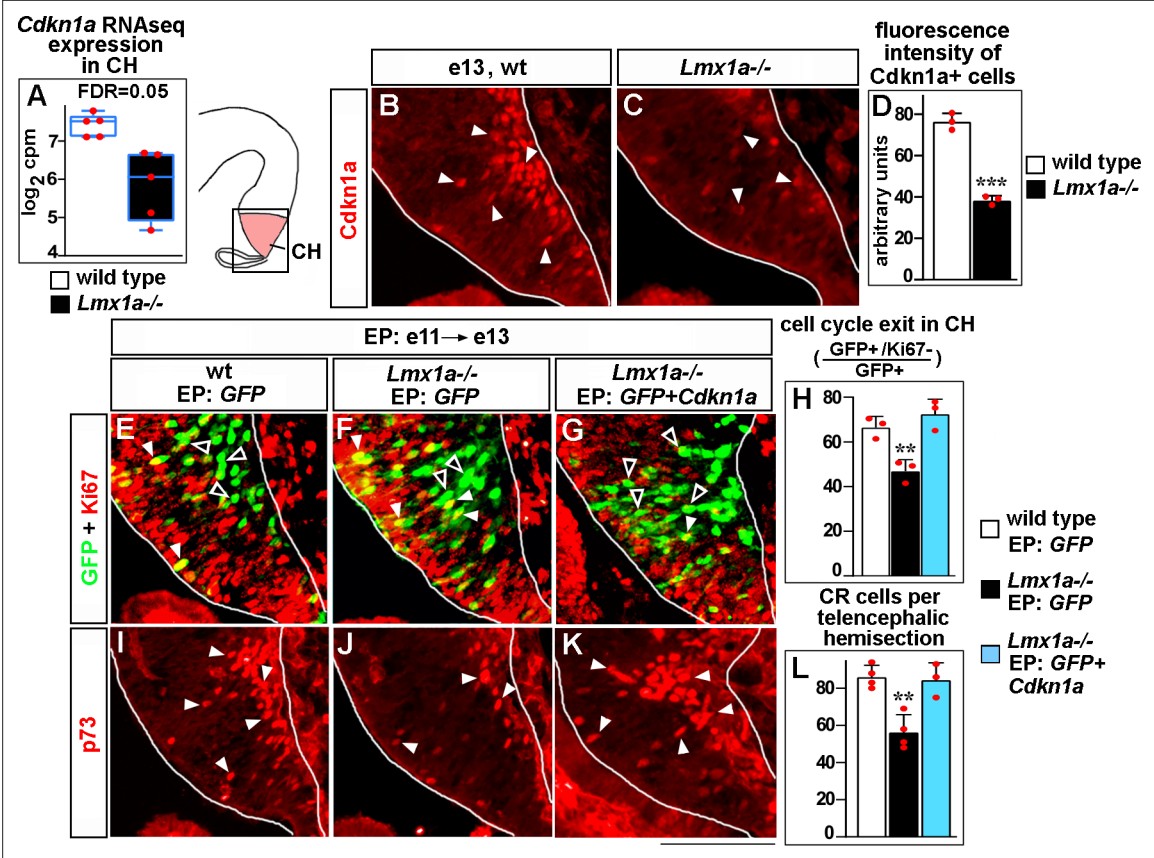

**Figure 5.** *Lmx1a* regulates exit of progenitors from the cell cycle and Cajal–Retzius (CR) cell differentiation in the cortical hem (CH) via *Cdkn1a*. (**A**) Normalized read counts for *Cdkn1a* from the RNAseq experiment (**Figure 2G and H**). (**B–D**) Arrowheads indicate Cdkn1a+ cells in the CH (**B, C**), which exhibited lower immunofluorescence intensity in e13 *Lmx1a*⁻/⁻ embryos (**B–D**). ***p<0.001, two-tailed *t*-test, n = 3 embryos per genotype. (**E–L**) Embryos were in utero electroporated (EP) with indicated genes at e11 and analyzed at e13. Sections in panels (**E**), (**F**), and (**G**) are adjacent to those in panels (**I**), (**J**), and (**K**), respectively. (**E–G**) Arrowheads indicate electroporated cells that re-entered the cell cycle (GFP+/Ki67+ cells, yellow); open arrowheads indicate electroporated cells that exited the cell cycle (GFP+/Ki67- cells, green). The reduced exit of progenitors from the cell cycle in *Lmx1a*⁻/⁻ CH was rescued by electroporation of *Cdkn1a* (**H**). **p<0.01 versus *GFP* electroporated wild-type and *Cdkn1a+GFP* electroporated *Lmx1a*⁻/⁻ embryos, one-way ANOVA with Tukey's post hoc test, n = 3 embryos per condition. (**I–L**) Arrowheads indicate p73+ CR cells. The reduced number of CR cells in *Lmx1a*⁻/⁻ mutants was rescued by electroporation of *Cdkn1a* (**L**). **p<0.01 versus *GFP* electroporated wild-type and *Cdkn1a+GFP* electroporated *Lmx1a*⁻/⁻ embryos, one-way ANOVA with Tukey's post hoc test, n = 3–4 embryos per condition. Scale bar: 100 μm.

The online version of this article includes the following source data and figure supplement(s) for figure 5:

**Source data 1.** Data points for **Figure 5D, H and L**.

**Figure supplement 1.** In utero electroporation targets progenitors in the cortical hem (CH) in e11 embryos.

**Figure supplement 1—source data 1.** Data points for **Figure 5—figure supplement 1D**.

coordinated to achieve the proper formation and function of specific signaling centers (**Bielen et al., 2017**; **Cavodeassi and Houart, 2012**; **Manfrin et al., 2019**; **Subramanian and Tole, 2009b**). Here we show that *Lmx1a* acts as a master regulator of the CH, a major signaling center that controls the formation of the hippocampus in the medial telencephalon (**Caronia-Brown et al., 2014**; **Mangale et al., 2008**; **Moore and Iulianella, 2021**). Our loss-of-function experiments revealed that *Lmx1a* is required for the expression of a broad range of CH markers and Wnt signaling molecules, the proper exit of CH progenitors from the cell cycle, and differentiation and migration of CR cells. In complementary gain-of-function experiments, we found that *Lmx1a* is sufficient to induce CH features (*Wnt3a* and *Ccdc3* expression, CR cells, and downregulation of expression of the cortical selector gene *Lhx2*, involved in the CH/hippocampus boundary formation) (**Figure 7**). This study identifies *Lmx1a* as the first intrinsic molecule that specifies CH fate and, to our knowledge, provides the strongest example that the orchestration of signaling center development in the CNS utilizes master regulatory genes.

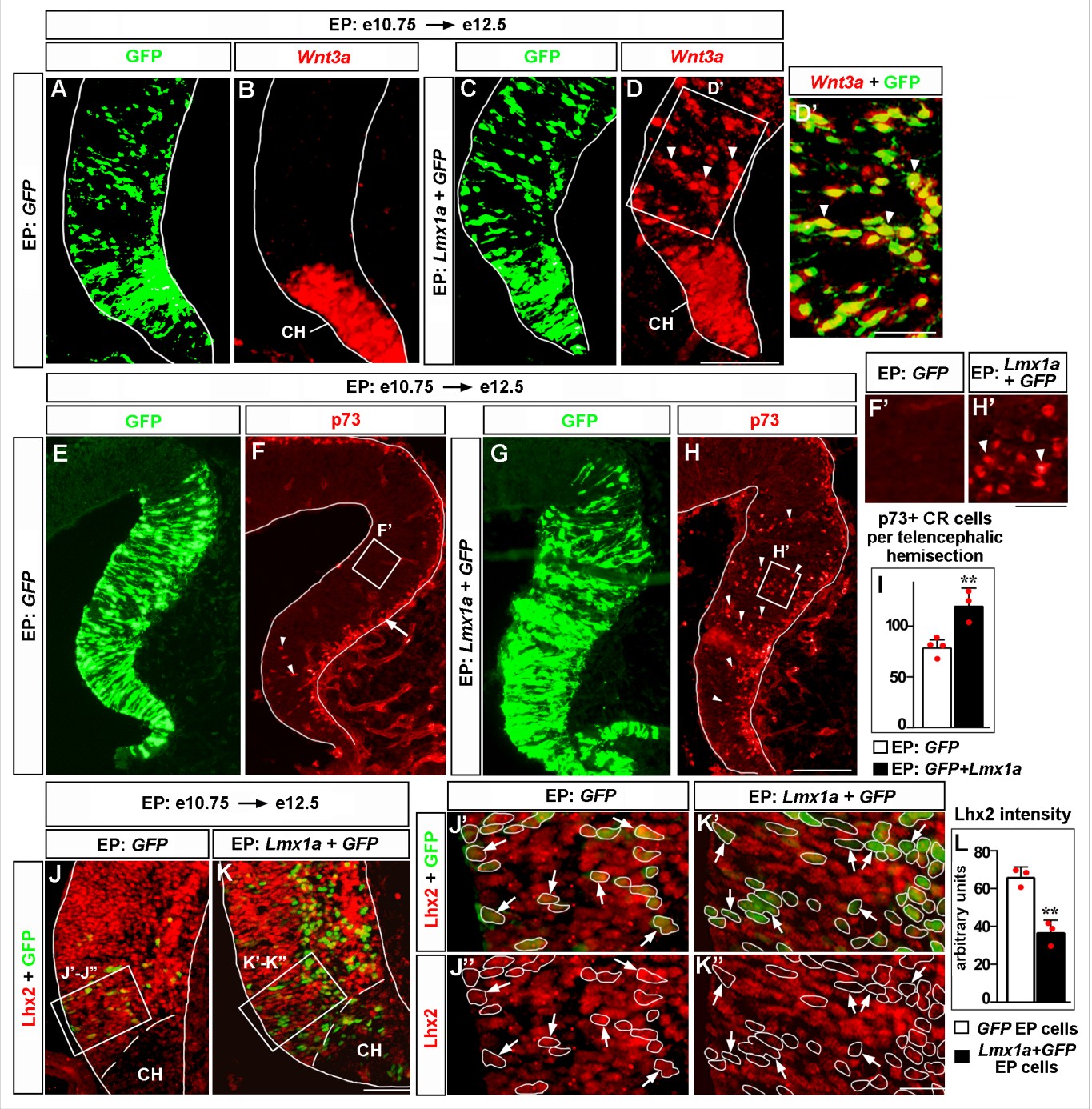

**Figure 6.** *Lmx1a* is sufficient to induce key features of the cortical hem (CH). Panels (**A**) and (**B**), (**C**) and (**D**), (**J′**) and (**J″**), (**K′**) and (**K″**) show the same sections imaged for different markers. Panels (**E–H**) show adjacent sections. wild-type embryos were in utero electroporated (EP) into the medial telencephalon (by placing electrodes as shown in *Figure 6—figure supplement 1*, top diagram) at e10.75 and analyzed at e12.5. (**A–D′**) In contrast to control (*GFP*-electroporated) embryos, in which *Wnt3a* expression was limited to the CH (**A, B**), in *Lmx1a*-electroporated embryos, ectopic *Wnt3a* expression was detected in the hippocampal field (arrowheads), clearly beyond the CH (**C, D**). Extensive overlap between GFP fluorescence and *Wnt3a* in situ hybridization signal (**D′**, arrowheads) suggests cell-autonomous induction of *Wnt3a* by *Lmx1a*. (**E–I**) In *GFP*-electroporated controls (**F, F′**), p73+ Cajal–Retzius (CR) cells were deeply located (arrowheads) only in the CH (where p73+ CR cells arise, *Siegenthaler and Miller, 2008*), while migrating superficially located p73+ CR cells (arrow) were found also in the cortical neuroepithelium. In *Lmx1a*-electroporated embryos, p73+ cells were found deeply located (arrowheads) throughout the medial telencephalic neuroepithelium (**H, H′**). The total number of p73+ CR cells increased in *Lmx1a*-electroporated embryos (**I**), **p<0.01, two-tailed *t*-test, n = 3–4 embryos per condition, further supporting induction of CR cells by *Lmx1a*. (**J–L**) *Lhx2* is expressed in the hippocampal primordium but not in the CH (**J, K**). Arrows indicate cells in the hippocampal primordium electroporated with *GFP*

*Figure 6 continued on next page*

*Figure 6 continued*

(controls) (**J′, J″**) or *Lmx1a+GFP* (**K′, K″**). The intensity of Lhx2 immunofluorescence was reduced in *Lmx1a*-electroporated cells (**J′–L**), indicating that *Lmx1a* is sufficient to repress *Lhx2* expression. **p<0.01, two-tailed *t*-test, n = 3 embryos per condition. Scale bars: 100 µm (**A–D, E–H**); 50 µm (**D′, J, K**); 30 µm (**F′, H′**); 15 µm (**J″–K″**).

The online version of this article includes the following source data and figure supplement(s) for figure 6:

**Source data 1.** Data points for *Figure 6I and L*.

**Figure supplement 1.** *Lmx1a* overexpression induces cortical hem (CH) marker *Ccdc3* but not choroid plexus marker Ttr.

**Figure supplement 2.** *Lmx1a* overexpression does not induce *Wnt3a* in lateral cortex.

Until now, no intrinsic molecule had been reported to induce CH. Although the loss of transcription factors *Gli3* and *Dmrt3/4/5* prevents CH formation, expression domains of all these genes expand beyond the CH, suggesting that they play permissive rather than instructive roles in CH development (*Grove et al., 1998*; *Kikkawa and Osumi, 2021*; *Quinn et al., 2009*; *Subramanian et al., 2009a*; *Subramanian and Tole, 2009b*). Furthermore, mosaic inactivation of the cortical selector gene *Lhx2*, prior to e10.5, resulted in the development of ectopic CH in *Lhx2*-null patches of the cortical neuroepithelium, suggesting that CH is the default fate in the medial telencephalon (*Mangale et al., 2008*). However, our overexpression of *Lmx1a* in the hippocampal primordium at e10.75, after the cortical neuroepithelium identity has already been assigned by *Lhx2* (*Mangale et al., 2008*), resulted in cell-autonomous ectopic expression of a CH signaling molecule Wnt3a, a CH marker *Ccdc3*, and the generation of ectopic p73+ CR cells, indicating that the CH identity is actively assigned by *Lmx1a*. Recently, *Lmx1a* was reported to activate a human CH-specific enhancer and an enhancer for the

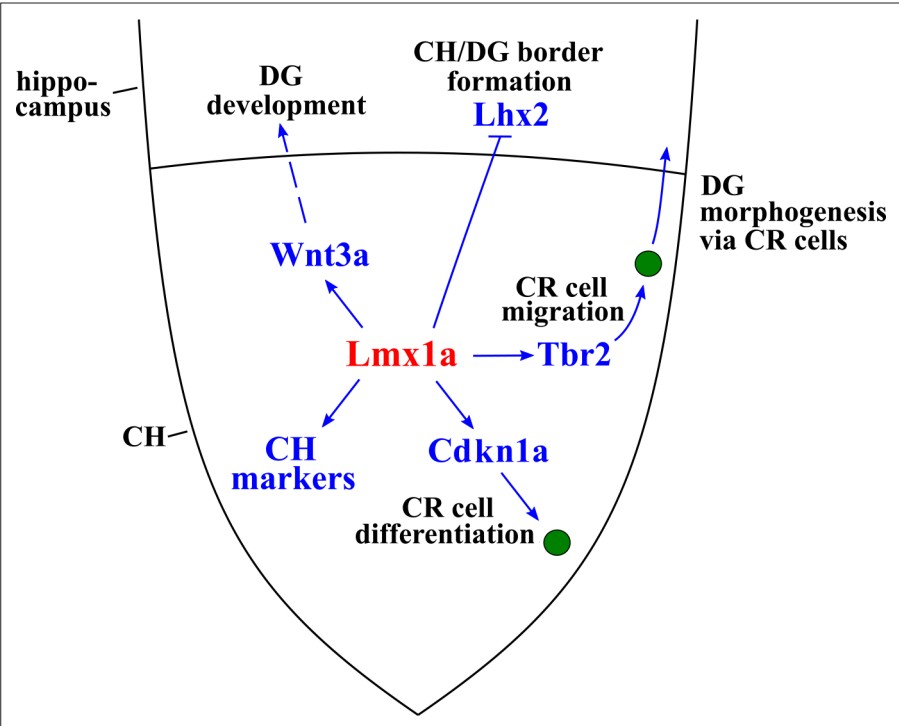

**Figure 7.** *Lmx1a*-dependent developmental processes and downstream mediators of *Lmx1a* activity in the cortical hem (CH). *Lmx1a* promotes expression (solid arrow) of secreted Wnt3a, which non-autonomously (dashed arrow) regulates the development of the hippocampal dentate gyrus (DG) (proliferation in the dentate neuroepithelium [DNe], the transhilar scaffold, and the number and input resistance of DG neurons). *Lmx1a* represses the expression of the cortical selector gene *Lhx2* (solid bar), segregating the CH from the adjacent hippocampal field. *Lmx1a* positively regulates the expression of a wide range of CH markers. It also promotes the exit of CH progenitors from the cell cycle and their differentiation into Cajal–Retzius (CR) cells by activating the expression of *Cdkn1a*, and positively regulates expression of *Tbr2* to promote migration of CR cells, which are necessary for the hippocampal fissure (HF) and transhilar scaffold formation.

mouse *Cux2*, the gene expressed in the CH and a subset of cortical neurons (*Fregoso et al., 2019*). Although the significance of the aforementioned enhancers for CH development or function remains unknown, these data are consistent with and further support the role for *Lmx1a* in the induction of the CH that we describe in this study. Interestingly, while *Lmx1a* induced CH features in the medial telencephalon, *Lmx1a* overexpression in the lateral cortex failed to induce ectopic expression of *Wnt3a*, indicating that medially expressed competence factors (permissive genes) are needed to maintain the CH-inducing activity of *Lmx1a*. Such factors are likely to include *Gli3* and *Dmrt3/4/5*, loss of which compromises the development of the endogenous CH (*Grove et al., 1998*; *Kikkawa and Osumi, 2021*; *Quinn et al., 2009*; *Subramanian et al., 2009a*; *Subramanian and Tole, 2009b*).

Although knockout/knockdown of *Lmx1a* dramatically reduces the expression of numerous CH markers (*Figure 2—figure supplements 3 and 4*), the CH was not completely lost in *Lmx1a-/-* mice (*Figure 3A–C*; *Chizhikov et al., 2010*), suggesting a partial compensation for the *Lmx1a* CH-inducing activity by another gene. Ablation of the Bmp signaling (in *BmpR1a/b* double knockout mice) was reported to prevent CH induction (*Fernandes et al., 2007*), while mice with downregulated Bmp signaling still develop a CH-like structure at the telencephalic dorsal midline (*Chizhikov et al., 2019*; *Hébert et al., 2002*). Moreover, while high levels of Bmps promoted the ChPe fate (*Doan et al., 2012*; *Watanabe et al., 2012*), low Bmp doses induced CH gene expression signature in embryonic stem cells in vitro (*Watanabe et al., 2016*). Thus, it is likely that an intrinsic molecule(s) activated by low levels of Bmp signaling acts partially redundant to *Lmx1a* to assign CH fate in the medial telencephalic neuroepithelium.

In addition to CH induction, we report that *Lmx1a* functions as a regulator of expression of CH-specific Wnt signaling molecules, including the canonical Wnt3a (*Figure 3A–C*, *Figure 2—figure supplement 4I*, and *Figure 3—figure supplement 1*). Canonical Wnt signaling from the CH promotes the proliferation of hippocampal (DNe) progenitors (*Lee et al., 2000*). It has also been previously shown that single and double mutants for *Lrp6* and *Lef1* genes, which encode components of the Wnt signaling transduction pathway, exhibit disrupted transhilar and supragranular scaffolds (*Zhou et al., 2004*; *Li and Pleasure, 2005*), indicating that Wnt signaling has a role in the development of the hippocampal glial scaffold. Our gene expression studies and phenotypic analysis of *Lmx1a-/-* mutant and *Lmx1a+/-;Wnt3a+/-* double heterozygous mice identified *Lmx1a* as a novel regulator of proliferation of DG progenitors, hippocampal glial scaffold formation, and electrophysiological properties (input resistance) of DG neurons, which likely, at least partially, promotes hippocampal development by modulating Wnt signaling, particularly expression of its secreted ligand Wnt3a.

We found that besides CH signaling, *Lmx1a* regulates two distinct steps of CR cell development – their migration and differentiation from CH progenitors. Migrating CR cells drive the formation of the HF and glial scaffold in the developing hippocampus, and *Tbr2* has been implicated in the regulation of migration of CR cells (*Frotscher et al., 2003*; *Hodge et al., 2013*; *Meyer et al., 2004*; *Meyer et al., 2019*). In addition to CR cells, *Tbr2* is also expressed in hippocampal progenitors (that do not arise from the CH) (*Hodge et al., 2013*; *Hodge et al., 2012*). Interestingly, in *Lmx1a* mutants, *Tbr2* expression was reduced in the CH, but not in hippocampal progenitors (*Figure 4B–D*). To study whether *Tbr2* intrinsically regulates CR cell migration, we conditionally inactivated this gene in the CH (in *Lmx1a-Cre;Tbr2F/F* mice), which resulted in CR cell migration abnormalities similar to those observed in *Lmx1a-/-* mice and *Lmx1a/Tbr2* double heterozygous (*Lmx1a+/-;Lmx1a-Cre;Tbr2+/F*) mice. In addition to CR migration abnormalities, the latter two mouse strains had similar HF and glial scaffold formation abnormalities. Taken together, our data indicate that in addition to regulating Wnt signaling to promote hippocampal development, *Lmx1a* also regulates *Tbr2* expression to promote CR cell migration and HF formation.

Initiation of CR cell migration coincides with their differentiation in the CH (*Causeret et al., 2021*). In addition to CR cell migration, neuronal differentiation was also disrupted in the *Lmx1a* mutant CH, as revealed by our RNAseq pathway analysis and a delayed exit of CH progenitors from the cell cycle, a prerequisite for cell differentiation (*Figure 2*). Interestingly, despite a delayed migration of CR cells, exit of progenitors from the cell cycle was normal in the CH of *Lmx1a+/-; Lmx1a-Cre;Tbr2+/F* mice (*Figure 4G–H*), indicating that CR cell differentiation and migration are regulated by *Lmx1a* via different downstream mediators. We found that loss or knockdown of *Lmx1a* in the CH reduces the expression of a negative regulator of the cell cycle *Cdkn1a* (*Xiao et al., 2020*) and that exogenous *Cdkn1a*, introduced via in utero electroporation, not only rescued cell-cycle exit defects in the CH but

also normalized the number of p73+ CR cells in *Lmx1a*$^{-/-}$ embryos. Together, our data indicate that *Lmx1a* promotes differentiation of CR cells at least partially via regulating the exit of their progenitors from the cell cycle via *Cdkn1a*.

In addition to the discussed above roles of *Lmx1a* in the regulation of proliferation in the DNe, transhilar scaffold formation, and differentiation and migration of CR cells, disruption of other cellular processes is likely to contribute to hippocampal deficits in *Lmx1a*$^{-/-}$ mice as well. For example, although no time-lapse imaging was performed, our BrdU labeling experiments revealed an abnormal distribution of cells originating in the DNe in *Lmx1a*$^{-/-}$ embryos (*Figure 1—figure supplement 2*), which in the absence of elevated apoptosis, suggests a compromised migration of cells from the DNe to DG. Although a reduced proliferation in the DNe of *Lmx1a*$^{-/-}$ mutants likely contributes to the reduction of their DG size, DNe proliferation abnormalities alone are unlikely to explain the relative change in the fractions of BrdU-labeled cells found in different segments of the migratory route in *Lmx1a*$^{-/-}$ embryos (a larger fraction of BrdU-labeled cells was detected in the DMS and a smaller fraction of BrdU-labeled cells was found in the FDJ and DJ regions, relative to wild-type controls). A severe disruption of the transhilar glial scaffold, the structure necessary for cellular migration from the DNe to the DG, is also consistent with the presence of migration abnormalities in *Lmx1a*$^{-/-}$ mutant mice.

Wnt3a, which is downregulated in the *Lmx1a*$^{-/-}$ CH, is known to promote not only proliferation but also the specification of DG progenitors (*Lee et al., 2000*; *Mangale et al., 2008*; *Subramanian and Tole, 2009b*). Thus, although not directly tested in the current study, it is likely that the reduced number of Prox1+ DG progenitors in *Lmx1a*$^{-/-}$ embryos results not only from their reduced proliferation but also because of their decreased specification. Finally, in *Lmx1a* mutants, we linked a decreased number of CR cells with a reduced exit of CH progenitors from the cell cycle. However, our data do not exclude a possibility that loss of *Lmx1a* also causes a cell cycle progression defect (resulting in CH progenitors being delayed in a certain phase of the cell cycle). This hypothesis remains to be tested.

Our current analysis has also strengthened the role of *Lmx1a* in the segregation of the CH from the cortical neuroepithelium. A boundary formation between the cortical (hippocampal) and CH neuroepithelia involves the cortical selector gene *Lhx2*. In the chimeric telencephalic neuroepithelium and in vitro cell aggregation studies, *Lhx2*$^{+/+}$ and *Lhx2*$^{-/-}$ cells self-segregate, forming a boundary between *Lhx2*$^{+/+}$ (*Lhx2* expressing) cells that adopt the hippocampal fate and *Lhx2*$^{-/-}$ cells that adopt the fate of the CH (*Mangale et al., 2008*). Our previous genetic fate mapping revealed that in the absence of *Lmx1a*, the dorsal midline lineage aberrantly contributes to the adjacent hippocampus (*Chizhikov et al., 2010*). Our current observation that *Lmx1a* is sufficient to downregulate *Lhx2* in e10.75–e12.5 cortical neuroepithelium (*Figure 6J–L*) indicates an intrinsic role for *Lmx1a* in the segregation of the CH lineage from cortical (hippocampal) cells. Interestingly, our RNAseq analysis revealed significant enrichment in cell–cell and cell–extracellular matrix adhesion pathways among genes misregulated in the *Lmx1a*$^{-/-}$ CH, suggesting that in addition to *Lhx2*, *Lmx1a* regulates CH/hippocampus segregation via adhesion genes.

In conclusion, we determined that *Lmx1a* orchestrates CH development and function by co-regulating multiple processes ranging from CH cell-fate induction to CH Wnt signaling to differentiation and migration of CR cells. We determined the genes misregulated by loss of *Lmx1a*, and, by combining gene expression, genetic analysis and in utero electroporation rescue experiments, identified *Wnt3a*, *Tbr2*, and *Cdkn1a* as key downstream mediators of *Lmx1a* function in the CH. It is likely that genes identified as misexpressed in *Lmx1a*$^{-/-}$ CH by our RNAseq analysis include additional important regulators of CH development or mediators of CH function, and further studies are required to fully characterize an *Lmx1a*-dependent network. Regardless, our work revealed that the development and function of the CH, a key signaling center in the mammalian brain, employs *Lmx1a* as a master regulatory gene and established the framework for characterizing the mechanisms that regulate the development and function of signaling centers in the CNS.

## Materials and methods
### Mice
Mice used in this study include *Lmx1a* null (*Lmx1a*$^{drJ}$, Jackson Laboratory strain #000636) (*Chizhikov et al., 2006*; *Deng et al., 2011*; *Kridsada et al., 2018*), *Wnt3a* null (Jackson Laboratory strain #004581) (*Takada et al., 1994*), *Tbr2*$^{floxed}$ (Jackson Laboratory strain #017293) (*Zhu et al., 2010*), and *Lmx1a-Cre*

(BAC transgenic mice that express a GFP-tagged Cre protein under the control of the *Lmx1a* regulatory elements, referred to as *Lmx1a-Cre* throughout the article) (*Chizhikov et al., 2010*). Animals were maintained on a mixed genetic background comprising C57Bl6, FVB, and CD1. Genomic DNA obtained from mouse tails was used for genotyping (*Chizhikov et al., 2019*; *Takada et al., 1994*; *Zhu et al., 2010*). Both males and females were used for analysis. The presence of a vaginal plug was considered as embryonic day 0.5 (e0.5), while the day of birth was considered postnatal day 0 (P0). For cell proliferation, cell cycle exit, and cell migration experiments, pregnant dams were intraperitoneally injected with BrdU at 50 mg/kg. All animal experiments were approved by the University of Tennessee Health Science Center (UTHSC) Institutional Animal Care and Use Committee (IACUC) (protocols 21-0248.0, 18-037.0, 15-057.0).

## Immunohistochemistry

Embryonic heads or dissected brains were fixed in 4% paraformaldehyde (PFA) in phosphate-buffered saline (PBS) for 1.5–12 hr, washed three times in PBS (30 min each), equilibrated in 30% sucrose in PBS for 2 hr at 4°C, and embedded in optimum cutting temperature (OCT) compound (Tissue Tek). Tissue blocks were sectioned coronally on a cryostat at 12 μm, and sections were collected on Superfrost Plus slides (Fisher Scientific). Immunohistochemistry was performed as described (*Iskusnykh et al., 2021*). Sections were dried for 20 min, washed three times in PBS (10 min each), blocked in PBS containing 1% normal goat serum and 0.1% Triton X-100, and incubated with primary antibodies at 4°C overnight. Slides were washed three times in PBS (10 min each), incubated with secondary antibodies for 1 hr at room temperature, washed three times in PBS (10 min each), mounted in Fluorogel (EMS), and cover slips were applied. For immunostaining with anti-Ki67, anti-BrdU, and chicken anti-Tbr2 antibodies, slides were boiled in 1× target retrieval solution (Dako) for 10 min and cooled at room temperature for 1 hr prior to the blocking step. We used the following primary antibodies: anti-Prox1 (rabbit, Chemicon, Cat# Ab5475, 1:500, RRID:AB_177485), anti-Prox1 (mouse, Novus, Cat# NBP1-30045, 1:200, RRID:AB_2170710), anti-BrdU (rat, Abcam, Cat# ab6326, 1:50, RRID:AB_305426), anti-BrdU (rabbit, Rockland, Cat# 600-401c29, 1:300, RRID:AB_10893609), anti-Ki67 (mouse, BD Pharmingen, Cat# 556003, 1:250, RRID:AB_396287), anti-GFAP (rabbit, Dako, Cat# z0334, 1:200, RRID:AB_10013382), anti-Reelin (mouse, Chemicon, Cat# Mab5364, 1:500, RRID:AB_2179313), anti-Tbr2 (rat, eBioscience, Cat# 14-487582, 1:200, RRID:AB_11042577), anti-Tbr2 (chicken, Millipore, Cat# AB15894, 1:200, RRID:AB_10615604), anti-GFP (chicken, Abcam, Cat# ab13970, 1:500, RRID:AB_300798), anti-p73 (mouse, Invitrogen, Cat# MA5-14117, 1:100, RRID:AB_10987160), anti-Lhx2 (rabbit, Invitrogen, Cat# PA5-64870, 1:200, RRID:AB_2662923), anti-Cdkn1a (mouse, BD Pharmingen, Cat# 556431, 1:500, RRID:AB_396415), anti-Ctip2 (rat, Abcam, Cat# ab18465, 1:500, RRID:AB_2064130), anti-Lmx1a (goat, Santa-Cruz, Cat# sc-54273, 1:500, RRID:AB_2136830), anti-Ttr (rabbit, Dako, Cat# A0002, 1:200, RRID:AB_2335696), and anti-activated Caspase 3 (rabbit, Promega, Cat# G7481, 1:250, RRID:AB_430875), and appropriate secondary antibodies, conjugated with Alexa 488 or 594 fluorophores (Life Technologies) at 1:100 dilutions.

## Histology

Tissue was processed as described for immunohistochemistry above. Slides were incubated in 0.1% cresyl violet acetate solution for 15 min, dehydrated in ethanol and xylenes, dried at room temperature, mounted in Permount (Fisher), and cover slips were applied.

## RNAscope fluorescent in situ hybridization

Embryos were fixed in 4% PFA for 16 hr at 4°C, submerged in 10, 20, and 30% sucrose, embedded in OCT, and sectioned coronally on a cryostat at 12 μm. RNAscope in situ hybridization was performed with RNA probes for the mouse *Wnt3a* gene (NM_009522.2, 20 ZZ pairs, target location: 667–1634) and the mouse *Ccdc3* gene (NM_028804.1, 9 ZZ pairs, target location: 2–702) and the RNAscope 2.5 HD Reagent Kit-RED (Advanced Cell Diagnostics, USA). Slides were boiled in 1× Target retrieval solution for 5 min, treated with protease for 15 min at 40°C, and incubated with the RNA probe for 2 hr at 40°C. The unbound probe was removed by rinsing slides in 1× wash buffer, and signal amplification reagents were sequentially added. Finally, slides were washed twice in PBS, mounted in Fluoro-Gel medium, and cover slips were applied (Electron Microscopy Sciences, USA). When slides from in utero

electroporated embryos were used, GFP fluorescence in each section was imaged to identify electroporated cells prior to the beginning of in situ hybridization.

## Laser capture microdissection (LCM) and RNA purification

LCM was performed using an Arcturus XT LCM machine. For RNAseq experiments (*Figure 2G and H*), the CH was isolated under a bright light mode. For *Lmx1a* knockdown experiments, achieved via in utero electroporation, electroporated (GFP+) cells in the CH were isolated using a fluorescent mode (see experimental design in *Figure 2—figure supplement 4A–D*).

For both types of experiments, embryonic heads were placed in OCT and flash-frozen on dry ice. Serial 10-μm-thick coronal sections spanning the entire telencephalon were collected. Every second section was immunostained against Lmx1a to confirm the presence of the CH and identify its boundaries. The remaining slides were used to laser capture microdissect the entire CH (for RNAseq experiments) or GFP+ (electroporated) cells in the CH (for *Lmx1a* shRNA knockdown in utero electroporation experiments). CH tissue isolated from different sections of the same embryo was pooled and comprised one sample for RNAseq or qRT-PCR analysis. In *Lmx1a* knockdown experiments aimed to confirm an intrinsic role of *Lmx1a* in the CH, only embryos showing GFP fluorescence in the CH but lacking that in the ChPe (where Lmx1a is expressed in addition to CH [*Chizhikov et al., 2019*] were used for analysis) (*Figure 2—figure supplement 4B*).

A Pico Pure RNA purification kit (Arcturus) was utilized to isolate RNA from laser capture microdissected samples, according to the manufacturer's instructions. RNAse-free DNAse I (QIAGEN) was used to remove traces of genomic DNA.

## RNAseq and bioinformatics analysis

All RNA samples were subject to quality control using a Bioanalyzer. Mean RNA integrity number (RIN) of the samples used for library construction and sequencing was 8.43 ± 0.69 (SD).

FastQC-v0.11.8 was used to perform quality control of the RNAseq reads. Remaining adapters and low-quality bases were trimmed by Trimmomatic-v 0.36 (*Bolger et al., 2014*) with parameters: Adapters:2:30:10 SLIDINGWINDOW:4:15 MINLEN:30. Paired-end RNA-sequencing data was aligned to the mouse genome (GRCm38) with gene annotations from Ensembl (v102) (*Aken et al., 2016*) using STAR-v2.7.1a with default parameters (*Dobin et al., 2013*). Read count per gene was calculated using HTSeq-v0.11.2 (*Anders et al., 2015*). Genes were filtered by requiring at least 10 read counts in at least three samples and then processed using EdgeR-v3.30.3 (*Robinson et al., 2010*) for normalized read count per million (CPM) and differential expression analysis. p-Values were adjusted by the FDR. Significant differentially expressed genes (DEGs) were declared at FDR < 0.1. Functional enrichment was done using R package clusterProfiler-v 3.18 (*Yu et al., 2012*) based on the biological

**Table 1.** Sequences of primer used in qRT-PCR.

| Primers | Reference |
| --- | --- |
| Slc17a8 F: ggtgtggggaccctctctgg<br>Slc17a8 R: cccagaagcgaagaccccgt | This study |
| Ccdc3 F: tatgccaaggtgctggcgct<br>Ccdc3 R: taaggttgagccgggagccg | This study |
| Adamts1 F: ctggcacctccggtggctta<br>Adamts1 R: gtccccatggtccccagctt | This study |
| Tbr2 F: ctacgggccatacgccggaa<br>Tbr2 R: gtagtgggcggtggggttga | This study |
| Cux2 F: ccctgaggaagaccccctcgg<br>Cux2 R: ccttggcccatcaggaccca | This study |
| Cdkn1a F: ggtcccgtggacagtgagca<br>Cdkn1a R: gggacccagggctcaggtag | This study |
| Wnt3a F: ctcctctcggatacctcttagtg<br>Wnt3a R: gcatgatctccacgtagttcctg | *Watanabe et al., 2016* |
| Gapdh F: cgacttcaacagcaactcccactcttcc<br>Gapdh R: tgggtggtccagggtttcttactcctt | *Liu et al., 2009* |

processes GO, hallmark, and curated gene set definitions from MSigDB (v7.4.1) (*Subramanian et al., 2005*).

## qRT-PCR analysis

cDNA was synthesized with the cDNA synthesis kit (Bio-Rad, Cat# 1708890). qRT-PCR was performed using a Roche LC480 Real-time PCR machine and SYBR Fast qPCR master mix (Kapa Biosystems) as described (*Iskusnykh et al., 2021*). Gene expression was normalized to that of the reference gene *Gapdh*. All samples were tested in triplicate using the primers listed in *Table 1*.

## Mouse in utero electroporation

For overexpression studies, cDNAs encoding full-length mouse *Lmx1a* and *Cdkn1a* were cloned into a *pCIG* plasmid (*Megason and McMahon, 2002*). For *Lmx1a* knockdown experiments, we used the previously characterized *Lmx1a*-targeting shRNA construct (*pLKO.1*-mouse *Lmx1a* shRNA, Sigma, TRCN0000433282) (*Fregoso et al., 2019*). The *pLKO.1*-non-mammalian shRNA construct (Sigma, SHC002) was used as a control. Plasmids were co-electroporated with *pCAG-GFP* (*Matsuda and Cepko, 2004*) to visualize electroporated cells.

Using a fine glass microcapillary, plasmid DNA (1 µg/µl for each plasmid) was injected into the lateral ventricles of the embryos. Electroporations were performed using paddle electrodes and a BTX ECM830 electroporator. The positive electrode was placed against the medial telencephalic neuroepithelium to target the CH or neighboring hippocampal primordium (as shown in *Figure 2—figure supplement 4A*) or against lateral cortex to target lateral telencephalon (as shown in *Figure 6—figure supplement 2*). Following electroporation, the uterus was placed back into the abdomen to allow continuing development. Upon harvesting, embryos were evaluated under a stereomicroscope for GFP fluorescence in the telencephalon. The telencephalon of GFP+ embryos was serially coronally sectioned, and those demonstrating GFP fluorescence in desired telencephalic areas were used for analysis.

## Electrophysiological recordings

Brains were dissected in artificial cerebrospinal fluid (aCSF), which contained 20 mM D-glucose, 0.45 mM ascorbic acid, 1 mM $MgSO_4$, 2.5 mM KCl, 26 mM $NaHCO_3$, 1.25 mM $NaH_2PO_4$, 125 mM NaCl, and 2 mM $CaCl_2$, and were cut into 300-µm-thick slices using a Leica VT1000S vibratome at 4°C. Slices were allowed to recover at 32°C for 60 min in an aCSF-filled chamber under continuous bubbling with 95% $O_2$/5% $CO_2$. For the recording, slices were put in a chamber attached to a modified stage of an Olympus BX51WI upright microscope and perfused continuously with bubbled aCSF at room temperature. Electrodes (4–8 MΩ) were pulled from borosilicate glass with an outer diameter of 1.5 mm using a P-97 micropipette puller (Sutter Instruments, USA) and filled with a solution containing 140 mM K-gluconic acid, 10 mM HEPES, 1 mM $CaCl_2$, 10 mM EGTA, 2 mM $MgCl_2$, and 4 mM $Na_2ATP$ (pH 7.2, 295 mOsm). Whole-cell current clamp recordings were obtained from neurons in the granule cell layer of the DG using an Axon Multiclamp 700B amplifier (Molecular Devices, USA). Traces were digitized using an Axon 1440A Digitizer at 10 kHz and filtered at 2 kHz using Clampex 10 software (Molecular Devices). After achieving a stable whole-cell recording, a multi-sweep current injection step protocol (−120 to 20 pA in 20 pA increments) was applied, and input resistance was calculated using Clampfit 9 software (Molecular Devices).

## Measurements, cell counts, and statistical analysis

Serial coronal sections were generated for each embryonic or postnatal telencephalon. The same number of sections was collected on each slide to facilitate the identification of the position of each section along the anterior-posterior (A-P) axis of the telencephalon. Upon inspection under a bright-field microscope, sections containing the CH (or its derivative – the fimbria in late embryonic and postnatal brains) were identified. For consistency, we compared sections from control and experimental animals taken at similar A-P levels, approximately midway along the telencephalic region that contained the CH/fimbria. At least three embryos were analyzed per experimental condition (typically 3–5 non-adjacent sections per animal).

Sections containing a fluorescent signal were evaluated under a Zeiss Axio Imager A2 fluorescence microscope and imaged using an AxioCam Mrm camera and AxioVision Rel 4.9 software

(Zeiss). Bright-field images were taken using an Olympus SZX microscope, Leica MC170HD camera, and LAS V4.5 software. In all experiments, slides for control and experimental animals were processed and stained in parallel and imaged using identical acquisition settings. Signal intensity (average pixel intensity) was measured in ImageJ software (NIH), as previously described (*Chizhikov et al., 2019*). Wnt3a RNAscope in situ hybridization signal was quantified in the entire CH (*Figure 3C*). To quantify Tbr2 expression, the CH was divided into three equally sized bins along the radial axis: ventricular zone, intermediate zone, and marginal zone (*Figure 4B*). Then immunohistochemistry signal for Tbr2 was quantified in all positive cells located in each of these CH regions (*Figure 4C*) and in a 200 μm high segment of the hippocampal primordium located directly above the CH (*Figure 4D*). Immunohistochemistry signal for Cdkn1a was quantified in all positive cells located in the CH (*Figure 5D*). Immunohistochemistry signal for Lhx2 was measured in all GFP+ cells in a 200 μm high segment of the hippocampal primordium located directly above the CH (*Figure 6L*). The signal intensity in individual cells was averaged to generate a value for each embryo. For these and other experiments, the boundaries of the CH were identified using adjacent Lmx1a-immunostained sections as templates (taking advantage that in the *dreher^J* strain used in this study, *Lmx1a* harbors a point mutation, which although completely functionally inactivates Lmx1a [*Chizhikov et al., 2010*; *Deng et al., 2011*] does not prevent the production of [inactive] Lmx1a protein).

To analyze proliferation, a 1.5 hr BrdU pulse (which labels cells in the S phase of the cell cycle) was used, and the percentage of BrdU+ cells (the number of BrdU+ cells divided by the total number of cells in the region of interest) or the number of BrdU+ cells per ventricular surface length were calculated. The percentage of cells that exited the cell cycle (the number of BrdU+/Ki67- cells divided by the total number of BrdU+ cells in the region of interest) was calculated in embryos 24 hr after BrdU injection. For migration experiments, mice were injected with BrdU at e14.5, shortly before the DNe progenitors begin migration to the DG, and the distribution of migrated BrdU-labeled cells was analyzed at e16 and e18.5. The fractions of BrdU+ cells located in different segments along their migratory route were calculated (*Figure 1—figure supplement 2C–H*): the first segment (labeled as DMS) was between the DNe and FDJ, the second was along the surface of the FDJ, and the third was in the DG, as previously described (*Cai et al., 2018*).

To evaluate apoptosis in the developing hippocampus, the number of activated Caspase 3+ cells was quantified per mm² of the embryonic hippocampal primordium (defined as the region located between the fimbria and the HF, *Figure 1—figure supplement 3A–F*) or as a % of activated Caspase 3+ cells among DAPI+ cells in the DNe (*Figure 1R–T*) or as a % of activated Caspase 3+ cells among Prox1+ cells in the DG (*Figure 1—figure supplement 3D', E', G, H–M*).

Comparisons of two groups were performed with an unpaired two-tailed *t*-test. Comparisons of multiple groups were performed with one-way ANOVA with Tukey's post hoc test. IBM SPSS Statistics (version 27) and Excel software were used. Quantitative data in all figures, except *Figures 4A and 5A*, and *Figure 2—figure supplement 3C*, and *Figure 3—figure supplement 1B*, which show RNAseq expression data in a log scale, are presented as the mean ± SD. In the aforementioned figures, data are presented as the median (center line), the first and third quartiles (upper and lower edges of the box, respectively), and error bars indicating the minimum and maximum data points. Graphs were generated with GraphPad Prism 6 (GraphPad) software.

## Materials availability

All unique/stable reagents generated in this study are available from the corresponding authors without restriction.

## Acknowledgements

We thank W Armstrong, M Ennis, R Foehring, T Ishrat, D Guan, and L Wang for advice on experimental protocols or sharing equipment, S Krat, L Mukhametzyanova, A Zakharova, and I Niess (UTHSC) for genotyping, and K Johnson-Moore and the UTHSC Office of Scientific Writing for manuscript editing. This work was supported by NIH R01 NS093009 and R01 NS127973 to VVC and UTHSC Neuroscience Institute.

## Additional information

### Funding

| Funder | Grant reference number | Author |
|---|---|---|
| National Institute of Neurological Disorders and Stroke | R01 NS093009 | Victor V Chizhikov |
| National Institute of Neurological Disorders and Stroke | R01 NS127973 | Victor V Chizhikov |
| UTHSC Neuroscience Institute | | Igor Y Iskusnykh Nikolai Fattakhov Victor V Chizhikov |

The funders had no role in study design, data collection and interpretation, or the decision to submit the work for publication.

### Author contributions

Igor Y Iskusnykh, Nikolai Fattakhov, Conceptualization, Formal analysis, Investigation, Methodology, Writing - original draft; Yiran Li, Data curation, Formal analysis, Investigation; Laure Bihannic, Project administration; Matthew K Kirchner, Investigation, Project administration; Ekaterina Y Steshina, Formal analysis, Investigation, Writing - review and editing; Paul A Northcott, Data curation, Formal analysis, Supervision, Investigation; Victor V Chizhikov, Conceptualization, Formal analysis, Supervision, Funding acquisition, Investigation, Methodology, Writing - original draft, Writing - review and editing

### Author ORCIDs

Igor Y Iskusnykh ⓘ http://orcid.org/0000-0002-1207-6075
Nikolai Fattakhov ⓘ http://orcid.org/0000-0001-6707-9727
Matthew K Kirchner ⓘ http://orcid.org/0000-0001-6960-7685
Victor V Chizhikov ⓘ http://orcid.org/0000-0003-2338-1267

### Ethics

This study was performed in accordance with the recommendations in the Guide for the Care and Use of Laboratory Animals of the National Institutes of Health and approved by IACUC of the University of Tennessee Health Science Center (protocols 21-0248.0, 18-037.0, 15-057.0). In utero electroporation was performed under isoflurane anesthesia, and every effort was made to minimize suffering.

### Decision letter and Author response

Decision letter https://doi.org/10.7554/eLife.84095.sa1
Author response https://doi.org/10.7554/eLife.84095.sa2

## Additional files

### Supplementary files
• MDAR checklist

### Data availability

RNA-seq data have been deposited in the NCBI Gene Expression Omnibus under the accession code GSE216165. All other data are included in the article or supporting information. Source data files have been provided for Figures 1-6 and relevant Figure supplements.

The following dataset was generated:

| Author(s) | Year | Dataset title | Dataset URL | Database and Identifier |
|---|---|---|---|---|
| Iskusnykh IY, Chizhikov VV, Li Y, Bihannic L, Northcott PA | 2022 | Lmx1a is a master regulator of the cortical hem | https://www.ncbi.nlm.nih.gov/geo/query/acc.cgi?acc=GSE216165 | NCBI Gene Expression Omnibus, GSE216165 |

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
