## [Editor Report]

This study clearly demonstrates the important role of Lmx1a in collaboration with Wnt signaling in the development of medial cortical structures, including the dentate gyrus. The studies elegantly show the role of Lmx1a in the development of the cortical hem and provide important new insights into the signals controlling formation of this structure. It also provides helpful context for previous studies in this area.

---

## [Decision Letter]

**Decision letter after peer review:**

Thank you for submitting your article "Lmx1a is a master regulator of the cortical hem" for consideration by *eLife*. Your article has been reviewed by 2 peer reviewers, and the evaluation has been overseen by a Reviewing Editor and Jonathan Cooper as the Senior Editor. The following individual involved in the review of your submission has agreed to reveal their identity: Astrid Deryckere (Reviewer #2).

Essential revisions:

As you will see there are a number of points raised by the reviewers. These are all fairly modest in scope and should be addressed either with additional data or textual revisions prior to resubmission. Please address all the comments of the reviewers in your response letter. The following issues will likely require additional data.

1) Please add images of Lmx1a+/- sections to figure 3A. Since this genotype is included in the quantification, an example should also be shown.

2) Please address point #4 raised by reviewer 1 with additional quantification of the Tbr1 staining.

3) Please add an evaluation of cell death markers to examine for alterations in cell death in the dentate of mutants to help distinguish between a cell death or a specification phenotype.

4) Both reviewers raised questions about the hem induction by ectopic Lmx1a expression. A few additional markers of choroid or hem would be useful to address this.

5) Reviewer 2 raises a valid point about the migration data and it would be appropriate to add additional data to address these possibilities or to modulate the discussion of the phenotype.

Other comments of the reviewers should be able to be addressed by textual revisions.

*Reviewer #2 (Recommendations for the authors):*

Specific comments:

Line 137-139: More analysis is required to show a migration phenotype (e.g. time-lapse imaging, or in vitro migration assay). Apoptosis has to be excluded. It would also be helpful to already describe the proliferation defect at this stage. Could the phenotype be solely due to proliferation (and maybe apoptosis)? It is clear that the fraction of BrdU cells is bigger in the DMS compared to the FDJ and DG, but is the absolute (or relative to the surface) number of proliferating cells also reduced in the mutant?

---

## [Author Response]

Essential revisions:As you will see there are a number of points raised by the reviewers. These are all fairly modest in scope and should be addressed either with additional data or textual revisions prior to resubmission. Please address all the comments of the reviewers in your response letter. The following issues will likely require additional data.1) Please add images of Lmx1a+/- sections to figure 3A. Since this genotype is included in the quantification, an example should also be shown.

In the original paper, we did not analyze/quantify *Lmx1a* expression in *Lmx1a^+/-^* embryos by in situ hybridization (we analyzed only wt and *Lmx1a^-/-^* embryos). There was mislabeling in the original Figure 3C, which should have read *Lmx1a^-/-^*, not *Lmx1a^+/^*. We apologize for this error and have corrected the typo (Figure 3C). In response to Reviewer 2 comment about a “limited robustness of quantification of in situ hybridization signal”, we isolated CH by LCM and analyzed *Lmx1a* expression by qRT-PCR. Since the above comment concerned the *Lmx1a^+/-^* genotype, we included not only wt and *Lmx1a^-/-^* mutants, but also *Lmx1a^+/-^* embryos (and *Wnt3a^+/-^*, and *Lmx1a^+/-^;Wnt3a^+/-^* embryos as well) (Figure 3D, E). These new data further support the role of Lmx1a in regulating *Wnt3a* expression, and all Wnt3a data are now properly presented.

For more detail, please see our response to the downregulation of Wnt3a expression comment of Reviewer 2.

2) Please address point #4 raised by reviewer 1 with additional quantification of the Tbr1 staining.

Since Reviewer 1 mentioned Tbr2, not Tbr1 (and we did not study Tbr1 in this paper) we assume that this request is related to Tbr2, not Tbr1. We performed additional quantification of Tbr2 as requested by Reviewer 1 (point #4). This analysis revealed that Lmx1a regulates Tbr2 expression across the bins in the CH, but most significantly (p<0.001) in the marginal layer of the CH.

3) Please add an evaluation of cell death markers to examine for alterations in cell death in the dentate of mutants to help distinguish between a cell death or a specification phenotype.

We performed a very detailed analysis of apoptosis at multiple stages (at e14.5, e16, e18.5, P3, and P21). No difference in apoptosis between wt and *Lmx1a^-/-^* embryos was found at any stage, indicating that misregulated apoptosis is not a significant contributor to the DG phenotype of *Lmx1a^-/-^* mutants. (Figure 1R-T; Figure 1 —figure supplement 3).

For more detail, please see our response to the first specific comment of Reviewer 2.

4) Both reviewers raised questions about the hem induction by ectopic Lmx1a expression. A few additional markers of choroid or hem would be useful to address this.

We now show that *Lmx1a* induces an additional CH marker *Ccdc3* in the medial telencephalon (Figure 6 – supplement 1A-D’) but not a choroid plexus marker Ttr (Figure 6 – supplement 1E-F’).

Also, we show that Lmx1a induces hem only in the medial, but not lateral telencephalon, since it does not induce *Wnt3a* expression in the lateral cortex (Figure 6 – supplement 2).

5) Reviewer 2 raises a valid point about the migration data and it would be appropriate to add additional data to address these possibilities or to modulate the discussion of the phenotype.

We added additional in vivo data related to the migration of cells from the DNe. First, we showed that BrdU pulse at e14.5 specifically labels progenitors in the VZ (DNe) (e.g., we defined an initially BrdU-labeled pre-migratory population). Then, in addition to the final (e18.5) stage, we added an intermediate (e16) stage to better assess distribution of BrdU-labeled cells from the DNe during development. These data are now shown in Figure 1 —figure supplement 2. Since we now show that apoptosis is not affected in *Lmx1a^-/-^* mutants at the analyzed stages (e16-e18.5) (Figure 1—figure supplement 3A-G), these data more strongly support the hypothesis of abnormal migration of cells from the DNe in *Lmx1a^-/-^* mice. We also modulated the discussion of the phenotype (page 22, lines 486-496).

For more detail, please see our response to the first specific comment of Reviewer 2.

Other comments of the reviewers should be able to be addressed by textual revisions.

In response to the request from Reviewer 2, we provide experimental evidence that our in utero electroporation targets progenitors in the CH (Figure 5 – supplement 1).

Textual revisions were also made to address other comments of the Reviewers (see our response to specific comments of each Reviewer).

Reviewer #2 (Recommendations for the authors):Specific comments:Line 137-139: More analysis is required to show a migration phenotype (e.g. time-lapse imaging, or in vitro migration assay). Apoptosis has to be excluded. It would also be helpful to already describe the proliferation defect at this stage. Could the phenotype be solely due to proliferation (and maybe apoptosis)? It is clear that the fraction of BrdU cells is bigger in the DMS compared to the FDJ and DG, but is the absolute (or relative to the surface) number of proliferating cells also reduced in the mutant?

As time-lapse imaging is not established in our lab, we performed additional in vivo migration analysis of BrdU-labeled cells (a well-established approach to study migration from the DNe in the developing hippocampus; for example, see Cai et al., (2018). Proc Natl Acad Sci U S A *115*, E2725-e2733). First, we showed that a BrdU pulse at e14.5 specifically labels progenitors in the VZ (DNe) (e.g., we defined an initially BrdU-labeled pre-migratory population). Then, in addition to the last (e18.5) stage, we added an intermediate (e16) stage to better assess distribution of BrdU-labeled cells from the DNe during development. These data are now described in Figure 1 —figure supplement 2 and the Result section (page 7, lines 143-154).

In addition, we performed a detailed analysis of apoptosis at the aforementioned developmental stages (e14.5, e16, 18.5, and also at later stages: P3, P21), and excluded elevated apoptosis as a major cause of cell distribution abnormalities (Figure 1- supplement 3, and page 7-8, lines 161-165). Taken together, these data now more strongly support the hypothesis of abnormal migration of cells from the DNe in *Lmx1a* mutants.

We analyzed proliferation at e14.5 with a short BrdU pulse and present a reduced proliferation in the DNe [a reduced fraction of BrdU^+^ cells and a reduced number of BrdU^+^ cells relative to the ventricular surface length (Figure 1N-Q)] before presenting our migration analysis, as requested by the Reviewer. We also report that consistently with the reduced number of Prox1+ cells in the DNe and reduced proliferation in the DNe, the total number of BrdU^+^ cells migrated from the DNe (those located in the DMS+FDJ+DG regions) is smaller in e18.5 *Lmx1a^-/-^* mutants compared to controls (Figure 1 – supplement 2I). However, a reduced proliferation in the DNe alone (detected at e14.5, when BrdU labeling for the migration experiment has occurred), is unlikely to cause a relative change in the fractions of labeled cells found in different segments of the migratory route of *Lmx1a^-/-^* embryos (a higher faction of BrdU-labeled cells in the DMS and lower fractions of BrdU-labeled cells in the FDJ and DG regions compared to controls.) A reduced proliferation in the DNe (assuming normal apoptosis and normal migration) would result in fewer migrated cells but correct relative proportions of migrated cells in the DMS, FDJ, and DG.

A suggested by the Editors, we also modulated the discussion of the possible migratory phenotype. In the Discussion, we state:

“For example, although no time-lapse imaging was performed, our BrdU labeling experiments revealed an abnormal distribution of cells originating in the DNe in *Lmx1a^-/-^* embryos (Figure 1 —figure supplement 2), which in the absence of elevated apoptosis, suggests a compromised migration of cells from the DNe to DG. Although a reduced proliferation in the DNe of *Lmx1a^-/-^* mutants likely contributes to the reduction of their DG size, DNe proliferation abnormalities alone are unlikely to explain the relative change in the fractions of BrdU-labeled cells found in different segments of the migratory route in *Lmx1a^-/-^* embryos (a larger fraction of BrdU-labeled cells was detected in the DMS and a smaller fraction of BrdU-labeled cells was found in the FDJ and DJ regions, relative to wild type controls). A severe disruption of the transhilar glial scaffold, the structure necessary for cellular migration from the DNe to the DG, is also consistent with the presence of migration abnormalities in *Lmx1a^-/-^* mutant mice.” (page 22, lines 486-496).

When we discuss abnormal distribution of BrdU-labeled cells originating from the DNe in *Lmx1a^-/-^* embryos, we now state that these data “suggest” (rather than indicate) an abnormal migration. (page 22, line 488).